# Development of an automated human scent olfactometer and its use to evaluate detection dog perception of human scent

Edgar O. Aviles-Rosa[1]*, Andrea C. Medrano[2], Ariela Cantu[2], Paola A. Prada-Tiedemann[2], Michele N. Maughan[3], Jenna D. Gadberry[4], Robin R. Greubel[5], Nathaniel J. Hall[1]*

1 Department of Animal & Food Science, Texas Tech University, Lubbock, Texas, United States of America, 2 Department of Environmental Toxicology, Forensic Analytical Chemistry and Odor Profiling Laboratory, Lubbock, Texas, United States of America, 3 Excet, Springfield, Virginia, United States of America, 4 Intrinsic24 LLC, Hayden, Idaho, United States of America, 5 K9Sensus Foundation, Lucas, Iowa, United States of America

* edgar.aviles-rosa@ttu.edu (EOA-R); Nathaniel.j.hall@ttu.edu (NJH)

**Data Availability Statement:** The data underlying the results presented in the study are available

## Abstract

Working Dogs have shown an extraordinary ability to utilize olfaction for victim recovery efforts. Although instrumental analysis has chemically characterized odor volatiles from various human biospecimens, it remains unclear what perceptually constitutes human scent (HS) for dogs. This may be in part due to the lack of methodology and equipment to train and evaluate HS perception. The aims of this research were 1) to develop an automated human scent olfactometer (AHSO) to present HS to dogs in a controlled setting and 2) use the AHSO to evaluate dogs' response to different scented articles and individual components of HS. A human volunteer was placed in a clear acrylic chamber and using a vacuum pump and computer-controlled valves, the headspace of this chamber was carried to one of three ports in a different room. Dogs were trained to search all three ports of the olfactometer and alert to the one containing HS. In Experiment 1 and 2, the AHSO was validated by testing two dogs naïve to HS (Experiment 1) and five certified Search and Rescue (SAR) teams naïve to the apparatus (Experiment 2). All dogs showed sensitivity and specificity to HS > 95% in the apparatus. In Experiment 3, we used a spontaneous generalization paradigm to evaluate generalization from the HS chamber to different scented articles exposed to the same volunteer and to a breath sample. Dogs' response rate to the different scented articles was < 10% but exceeded 40% for the breath sample. In Experiment 4, we replicated this result by re-testing spontaneous generalization to breath and when the volunteer had breath exhausted/removed from the chamber. Dogs' response rate to breath alone was 88% and only 50% when breath was removed. Altogether, the data indicate that exhaled breath is an important and salient component of HS under these conditions.

within the Supporting information files of the manuscript.

**Funding:** This research was funded by a contract from the US Army DEVCOM Chemical Biological Center and Excet Incorporated to Texas Tech University. The funders had no role in the study design, data collection and analysis, decision to publish, or preparation of the manuscript. The view and ideas expressed herein are those solely of the author and do not necessarily reflect the official view or position of the sponsor.

**Competing interests:** The authors have declared that no competing interests exist.

## Introduction

The first documented evidence of the use of dogs to find missing persons date back to the 17th century where a group of Augustine monks in the monastery and hospice of the St. Bernard Pass in the Alpine mountains used dogs to find missing travelers and find people during avalanches [1,2]. Subsequently, dogs were used in World War I to find wounded soldiers. Ambulance or Red Cross dogs caried first aid to the wounded soldiers in battle [2,3]. The Search and Rescue (SAR) team as we know it today originated after World War II. A dog SAR team is composed of a trained dog and their handler. SAR dogs can be divided into three main categories based on the way they are trained to locate the victim. Tracking dogs methodically follow human scent on the ground (also ground disturbance) by working close to a pathway, however, tracking dogs are not typically pre-scented [4]. Trailing dogs are trained to follow a scent plume which could be either air borne or settled on the ground/vegetation. The dog will use whichever technique will get them to the target the most efficiently. Trailing dogs are typically pre-scented on an object on each trailing instances [3,4]. These dogs are expected to provide rescue teams with the direction of travel of the victim. Air-scenting SAR dogs are trained to sniff the air and detect the presence of human scent [3]. These dogs navigate the environment until locating the scent of a missing person.

Human scent (HS) detection dogs are an essential and effective tool utilized by rescue teams all over the world [5–10]. Different studies in insects, rodents, and crustaceans, have revealed that animals use olfactory driven navigation mechanisms to locate food sources, mates, and to avoid predators [5,11]. Through operant conditioning, humans have modified dogs' natural ability to track biologically relevant odors (i.e., prey, mate etc.) to locate different odors including human scent in complex environments. However, to date little is known about what is the main constituent of the human scent that detection dogs are utilizing to locate a person, or whether all categories of SAR dogs use the same feature of HS to complete the task.

HS is a complex blend of volatile organic and inorganic compounds of skin, breath, and bodily fluids origin [12]. Different attempts have been made to characterize human scent utilizing different analytical techniques and sensors [12–21] but none of these attempts included the use of detection dogs. The odor dynamics of volatile compounds from a human body have been well studied [22], but few studies have tried to understand how detection dogs perceive HS. Although from an analytical perspective, scientists have identified an array of human associated volatile compound mixtures, to date, it is still unknown how these findings apply to detection dog perception. Thus, there is still a need for research to understand what is the main source of HS that detection dogs are utilizing. Therefore, the aims of this research are: 1) to develop an automated human scent olfactometer (AHSO) that can be utilized to evaluate HS detection dogs' performance in a controlled laboratory setting, 2) validate the apparatus with certified SAR dogs trained to detect HS in other contexts, and 3) utilize the AHSO to conduct olfactory testing to evaluate what is the main constituent of HS from a detection dog perspective (i.e., volatiles of breath or skin origin, etc.).

## Materials and methods

### Apparatus

We developed an Automated Human Scent Olfactometer (AHSO) by modifying the olfactometers previously described and validated by our team [23]. The apparatus was redesigned to present the headspace of a chamber containing a volunteer person as the human scent-source and two other identical chambers as distractor sources. The headspace of these odor sources

was carried by a vacuum pump into our previously developed olfactometers [23] allowing us to utilize a total of 8 distractors/testing odors during a training or testing session.

The human scent and distractor chambers were made of a clear acrylic box (81.21 x 81.28 x 91.44 cm; 603.5 m$^3$). The tops of the boxes were removable to allow access to the interior. The boxes were sealed with acrylic cement (Craftics® #33) and a rubber draft seal strip (Cloud Buyer Professional sealing solution) to reduce extraneous air intrusion and odor losses. Volunteers entered the chamber by climbing a ladder. The chamber was then closed to allow the headspace to build up. The chamber was not locked, allowing volunteers to open the chamber at any time during testing. An electronic sensor (MYWHITENG, air quality and CO2 sensor) was inside the chamber to monitor CO2 levels, temperature, and humidity within the chamber to ensure volunteer safety and comfort. An air pump provided each chamber with clean filtered air (~10 L/m). This provided volunteers with a constant surplus of fresh air. One chamber, the distractor chamber, included a replicate of all items within the volunteers' chamber (e.g., clothes, any electronic devices, etc.) with the exception of the volunteer. A second chamber, the blank chamber, served as background control for the chambers itself, and was otherwise empty.

Fig 1 shows a schematic of the AHSO. All three chambers (the target, distractor, and blank) contained identical components. Three independent air pumps introduced clean air to the chambers identically. Air from the pumps passed through a charcoal filter (Omnipure; K2533 JJ) before entering the chamber. An oilless 12V mini diaphragm vacuum pump was connected to the bottom of the chambers. This pump pulled air from the chambers into a Teflon (PTFE) 3-way relay valve (Clippard, NIV1). Before reaching the PTFE relay valve, the air from all chambers passed through an acrylic casing (Pentek, 158110). Humidity within the volunteer chamber sometimes exceeded 90% relative humidity during testing or training, and the acrylic casing was intended to condense and retain the water (e.g., moisture trap) in the air before entering the control system. Removing excess water from the air was necessary to prevent odor contamination, damage to olfactometer parts, and to reduce any perceived differences between the air coming from the volunteer and distractor chambers (e.g., humid air associated with the target vs dry air associated with distractors chambers and odors).

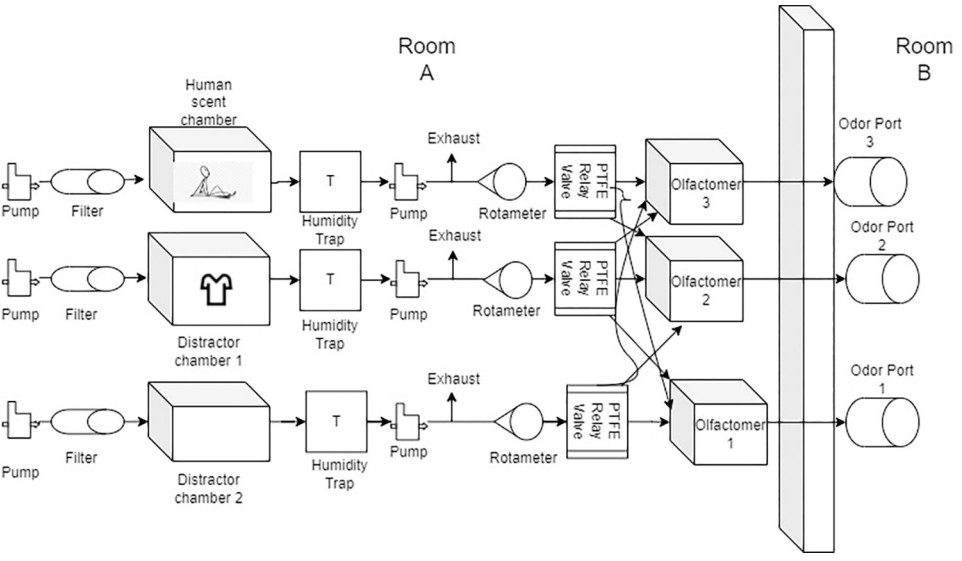

**Fig 1. Schematic of the Automated Human Scent Olfactometer (AHSO).**

Inline exhaust electronic valves regulated pressure in the system when the relay valves were closed. This was to prevent pressure from building up in the system and to ensure there was a constant flow of air in the chamber containing the volunteer. Inline rotameters regulated the airflow coming from the chambers to the olfactometers. Relay valves consisted of three independent PTFE valves connected to a common airline coming from a chamber. The activation of one of the valves directed the headspace of a chamber to one of the three olfactometers.

In addition to the acrylic chamber, the AHSO consisted of three independent olfactometers. These olfactometers were identical to the ones previously described by Aviles-Rosa et al. [23]. Each olfactometer was equipped with six additional solenoid valves that were connected to distractor vials. The activation of these distractor valves and of the relay valves were controlled by the olfactometer program. Once a valve was activated, the headspace of the chambers or of a distractor odor was carried to odor ports in an adjacent room for dogs to sample. The program was built to randomize the presentation of the headspace of the chamber with the volunteer and distractors (e.g., distractor chambers and distractors odors within the olfactometers) to three ports where dogs could sample the odors. For instance, during a trial where the target odor was present, one of the three olfactometers presented dogs with the headspace of the chamber with the volunteer while the other two olfactometers presented the dogs with 2 distractor odors selected randomly from the available options. The activation of one of the valves within the relay valves allowed the passage of the headspace of a chamber to a specified olfactometer. Within an olfactometer, the headspace from a chamber or a distractor jar (~2 L/m) was diluted with a continuous clean airline (~1L/m). The mixture (2:1 odor: dilution) was then presented to the dogs in an odor port in a separate room (Fig 1). This dilution ratio (e.g., 66% odor dilution) was selected to provide a moderate concentration of the chamber headspace. Each odor port was equipped with infrared beam-break sensors (IR) that measured if a dog sampled a port (e.g., introduced their nose in the port) and the duration of each beam break. Exhaust fans were at the top of each port, and between trials were activated to clear the odor from each port before the initiation of the next trial. Polyvinyl chloride (PVC) pipes carried the exhaust air to the outside of the room through a window. All odor whetted materials prior to introduction to the canine were made of glass, PTFE, stainless steel, or the acrylic chamber.

A series of experiments were conducted to validate the apparatus as a novel automated system to present HS to detection dogs. All animal testing and training procedures were reviewed and approved by the Texas Tech University Institutional Animal Care and Use Committee (Protocol # 2022–1158).

## Experiment 1: Training naïve dogs to human scent in the AHSO

**Participant dogs.**   Two mixed breed pet dogs (Stella- spayed female, 6y; and Buster- neutered male, 5y) were used for this experiment. These dogs were naïve to human scent detection but had prior training with odor detection in an olfactometer.

**Training.**   Dogs were trained to search all three ports of the olfactometer and alert to the port containing the headspace of the chamber with the human volunteer. This headspace should contain human specific volatiles of breath, skin, and body fluid origin (i.e., sweat) that are representative of whole-human scent (HS). As an alert, we trained dogs to hold their nose in the port for 3 or 4 s (Stella 3 s and Buster 4s). Stella and Buster were trained with differing alert times because due to the morphology of Stella's nose, she was not always activating the IR beam sensor during initial training, even when her nose was inside the port. The nose hold duration was recorded automatically by the IR sensors in front of the ports.

The beginning of a trial was marked with a trial initiation tone. This tone was used by the experimenter to know when odor was available in the ports and to give the search command

to the dogs. After the trial initiation tone, dogs had 45 s to search all three ports and make a response. If dogs alerted to the correct port, the computer program marked the response with a "bleep" sound and the experimenter reinforced the alert with a treat. In trials where HS was not present in any of the ports (e.g., blank trials), we trained the dogs to search all three ports and not alert to any of them for a period of 5 s (e.g., "all clear" response). During this 5 s period, dogs were trained to return and sit in front of the experimenter as an all-clear response. Correct all clears were also marked by the computer program with a "bleep" and were reinforced by the experimenter. False alerts (e.g., dog alerted to an odor port containing a distractor odor) and false all clear (e.g., an all-clear response when the target odor was present; odor miss) were marked by the computer with a "buzzer" sound to indicate to the experimenter that the response was wrong and were not reinforced. The end of a trial was also marked with a trial termination tone. This indicated to the experimenter that the exhaust fans were on. All tones used by the computer program were audible to the dog. The computer program also randomized and balanced the number of times the target odor appeared in each port. Training sessions consisted of 20 trials. Dogs received up to four training sessions in a day. The specific training parameters within a session changed based on the training phase (see below).

*Phase 1*: *Initial Training*. Dogs were given 20-trials sessions where the target odor was presented in all trials (100% odor prevalence). During this phase, the experimenter was not blind to the port containing the target odor. This facilitated training and shaping the dogs' alert. Initially the nose hold criterion was set at 1s and it was gradually increased. The final nose hold criterion was 4 and 3 seconds for Buster and Stella, respectively. Vinyl gloves (Med Pride), cotton balls, and a blank vial were used as distractor odors. Dogs received eight training sessions in this phase. The initial 4 sessions were with the same volunteer (P1-Adult Female) and the remaining sessions were with two different volunteers (P2- Adult Female; P3- Adult Male: 20–23 years). Between sessions, the chamber containing the human volunteer was cleaned with a 99% isopropyl alcohol solution (Florida Laboratories) and left open for at least 2 h. This was to remove any residual scent from the previous volunteer and ensure the dogs were alerting to the headspace produced by the new volunteer. The number of sessions dogs received with each volunteer as the target odor was dependent on volunteers' availability. Each volunteer was instructed not to shower, use perfume, deodorant, lotion, or anything with a strong odor for at least 4 hours before entering the chamber. Volunteers were allowed to use electronics and books to entertain themselves within the chamber during testing and training. A copy of the source of entertainment was placed in the distractor chamber (Phase 2 and onward) to control for any non-human extraneously introduced odorants.

*Phase 2*: *Distractor Training*. During Phase 2, the target odor frequency was reduced to 80% (e.g., 4 of the 20 trials within a session where blank trials) and all testing was conducted double-blind. From this Phase onward, all testing was conducted double-blind. All correct responses were reinforced by the experimenter with food. During this phase, the blank acrylic chamber and the distractor acrylic chamber were added as distractors. This served as a control to ensure dogs were alerting to the human specific volatiles within the headspace of the chamber and not to items within the chamber. In addition, volunteers were asked to bring personal hygiene items they use regularly (e.g., deodorant, shampoo, perfume, lotion etc.). These were also used as distractor odors in the olfactometer in addition to vinyl gloves, cotton balls, and a blank vial. Each volunteer had its own personal hygiene product (PH) as distractors within a session. To ensure equal air flow to all the olfactometers, the distractor and blank chamber were preprogrammed to appear in only one odor port within a trial. Because of this, the distractor odors were selected pseudo randomly by the computer program. During a trial that contained the target odor, first the computer program selected the port that will contain the headspace of the chamber with the volunteer. Then the program selected randomly a distractor

odor to be presented in the second port. If the distractor or blank box was selected, the computer removed it from the list and randomly selected a second distractor from the remaining distractors available. The same procedure also occurred during blank trials where the target odor was not presented to the dogs. Because of this randomization procedure, a distractor odor or chamber did not appear in every trial. The number of times a distractor odor appeared within a session was random.

Within a trial, the first response was recorded by the computer program, but after an incorrect response, dogs were allowed to search the olfactometer until making a correct response as part of the training. This was to facilitate training because we added novel distractors during this phase and all testing was conducted double-blind. The same volunteers from Phase 1 were used in Phase 2. Dogs received 9 training sessions in this phase.

*Negative Control Test.* At the end of the nineth session, dogs received a negative control test. The negative control test consisted of a 10-trial session with 80% target odor frequency identical to other sessions, but during the control test there was no volunteer in the chamber (e.g. the valve activations were identical to a regular session and scored identically). The control test was conducted to ensure dogs were alerting to the presence of human specific volatiles within the headspace of the chamber and not to any other unintentional cue from the olfactometer or the chamber (e.g., valve sounds, uncontrolled changes in air flow etc.). If the dogs were alerting exclusively to the HS, there would be a significant decrement in the proportion of correct responses in this control test. In contrast, if dogs could identify the target chamber without the presence of human scent at a rate greater than chance (e.g. 33%), this would identify that dogs can use cues other than human scent to identify the "correct" port. In this brief test, all procedures and air delivery/valve activation were identical except for the removal of the volunteer from the chamber and was used to ensure canine performance was controlled by human scent and not unintentional cues. It was expected that dogs would respond "all clear" on the control trials.

*Phase 3*: *Intermittent Schedule Training.* During this phase we gradually trained dogs to an intermittent schedule of reinforcement for correct responses following the method described by Aviles-Rosa et al. [24]. The reinforcement rate for correct alerts and correct all clears was initially set at 80%. Non-reinforced trials (probes) were selected randomly by the computer program and independent of dogs' responses, a probe trial terminated with no feedback (i.e., no correct or incorrect tone, only the trial termination tone) and no reinforcer was delivered. If dogs' performance under this reinforcement rate was > 80% correct responses, the reinforcement rate of correct all clears and correct responses was decreased to 50 and 70%, respectively. Thus, the overall reinforcement rate of correct responses within a session was 65%. Training dogs under an intermittent schedule of reinforcement was done to allow for spontaneous generalization assessments, and is standard procedure for generalization testing [e.g., 24–27]. In addition to the blank and distractor chambers, and the personal hygiene products of each volunteer, we introduced limonene ($10^{-3}$ v/v; CAS-5989-54-8) and isoamyl acetate ($10^{-3}$ v/v CAS-123-92-2) as novel distractors. The HS source for 3 of these sessions were the volunteers of previous phases, and for two sessions, the source of HS were novel volunteers (P4 and P5-Adult Female; 20–23 years).

*Validation test with novel volunteers.* After dogs' performance was > 80% correct responses under an intermittent schedule, we tested dogs' ability to transfer their training to identify the headspace of five novel volunteers (P6, P7, P8- Adult Males (21–35 years); P9 and P10- Adult females (20–23 years)). The purpose of this was to test if after training with just 5 volunteers, dogs were able to generalize to 5 novel volunteers or sources of HS. Testing parameters were as described in Phase 3. Each dog received one testing session with each volunteer. The chamber was cleaned between sessions with ethanol. We used the personal hygiene product of each

volunteer as novel distractors for each session. Thus, in each session, dogs encountered not just the scent of the novel volunteer but also novel distractors. This was to ensure that dogs were not alerting to a novel odor but rather were generalizing between volunteers. A second *negative control test* was conducted at the end of the testing phase.

**Variables measured.** For each session we calculated the proportion of correct responses (# correct responses/20), false alerts (# false alerts /20), and false all clears (# of false all clears/ 16 odor present trials). In addition, we calculated dog sensitivity and specificity during the validation test once they were considered fully trained. Sensitivity was defined as the number of times a dog correctly alerted to the port containing the headspace of the chamber with the volunteer after sampling the port, divided by the number of times a dog sampled a port containing the headspace of the chamber with the volunteer. Specificity was calculated by dividing the number of times a dog did not alert to a distractor odor after sampling the port, over the number of times the dog sampled a port with a distractor odor [28]. We utilized the IR readings to determine if a dog sampled a port and the number of distractor odors a dog sampled within a trial. In addition, for each session we calculated the false alert rate to each distractor odor. This was done by dividing the number of false alerts to a distractor over the total number of false alerts. Because we only had two participant dogs, no null hypothesis testing was conducted. The values reported are the mean ± the 95% confidence intervals of the different variables measured.

## Experiment 2: Validation of the AHSO with certified Search and Rescue (SAR) teams

Following the training of two naïve dogs to human associated odor in the AHSO in Experiment 1, we wanted to further confirm the odor presented by the AHSO was fully representative of human scent. To do this, we recruited a team of search and rescue canines that have been trained and certified independent of the research team, to evaluate whether dogs trained to find HS outside of the AHSO would respond to the odor output of the AHSO as being representative of HS.

**Participants.** Eight certified search and rescue (SAR) dog teams from a federal Task Force were the participants of this experiment. These teams were tested in two separate days (4 teams each day). Prior to this experiment, all dogs were certified either by the Federal Emergency Management Agency (FEMA) or the International Police Work Dog Association (IWPA) within the past 3 years. Most of the dogs were certified by both agencies. Table 1 shows additional information of each team. All teams traveled to the Texas Tech University Canine Research and Education Lab (CORE) for a single day of testing. The apparatus and settings were as described in Experiment 1.

**Table 1. Search and Rescue (SAR) team information.** Teams with the same superscript share the same handler.

|  | Years as a Handler | Dog Breed | Dog age | Years Certified | Years as Team |
|---|---|---|---|---|---|
| Team 1 | 20y | Labrador Retriever | 4y | 1y | 1.5y |
| Team 2 | 15y | Dutch Shepherd | 9y | 6y | 7y |
| Team 3* | 8y | German Shepherd | 7y | 5y | 5.5y |
| Team 4* | 8y | Belgian Malinois | 6y | 2.5y | 4y |
| Team 5 | 8y | Labrador Retriever | 9y | 7y | 8y |
| Team 6# | 24y | Golden Retriever | 10y | 8y | 10y |
| Team 7 | 3y | Mix | 6y | 1.5y | 3y |
| Team 8# | 24y | Golden Retriever | 3y | 0.5y | 3y |

**Training.**   Upon arrival to the lab, dog handlers were instructed about how the AHSO works and the purpose of the study. After a brief discussion, the experimenter answered any questions or concerns about the apparatus. Once the handler understood the concept and purpose of the study and signed informed consent, dogs started training to work the apparatus. Initial training consisted of 15- trials. These trials were split into two sessions each at a 100% target odor frequency. Gloves (Med Pride Nitrile Gloves), cotton balls, isoamyl acetate ($10^{-3}$ v/v CAS-123-92-2), the volunteer's deodorant, and the blank and distractor chambers were used as distractors for this experiment. Distractor odor randomization was the same as in Experiment 1.

During training, the experimenter was in the testing room with the handler and the dog. The experimenter was responsible for running the olfactometer program and ensuring the equipment was working properly. Neither the handler nor the experimenter were blind during initial training to facilitate handlers learning how the system works. After the trial initiation tone, the experimenter indicated to the handler which port contained the headspace of the chamber with the volunteer and each handler was instructed to train their dog to search the ports (introduce their nose in the port) of the apparatus and to alert to the port containing the target. Each handler trained their dog with positive reinforcement. Food and/or toys were used as reinforcers based on handlers' preference. The final trained alert for all the dogs was to bark facing at the port containing the HS. After the first 10-trial session, dogs received ~ 30-minute break. After this break, they completed the remaining five training trials. The purpose of these 15 training trials was to familiarize and train dogs and handlers to operate the AHSO. This initial training was done to ensure that dog teams knew the task and to confirm that their performance during testing in the AHSO was not affected by the novelty of the task.

The two training sessions were video recorded with the consent of the participants. The last 5-trial training session of each dog in the AHSO was coded by two observers to determine if dogs successfully acquired the AHSO task before testing. We considered the task acquired if the dog inserted its nose in the port containing the HS in at least four of the five trials in the training session. This criterion was established to ensure that dogs were accurately sampling the ports and thus were alerting to the odor within the port. For instance, if a dog did not sample the port containing the headspace of the target chamber, it would not be able to make an accurate response because HS is confined within the port. Video coders were blind to the port containing the HS when coding the videos. Video coding consisted of recording which of the three ports dogs sampled (introduced its nose). This data was subsequently merged with the AHSO data to determine if the dogs sampled the target port or not. We utilized video coding to determine if a dog searched a port rather than the IR reading from the AHSO directly because the IR readings were affected by tail wagging, and handler interference.

**Testing.**   Five of the eight dog teams met training criterion within 15 trials and received a single testing session of 10 trials with a target odor frequency of 80%. Testing sessions were conducted under single-blind conditions. Handlers were only instructed that the 10-trial session will include blank trials, and that for each trial, they will have to tell the experimenter to which port the dog alerted, or if the dog did not alert to any port (all clear). Both the handler and the experimenter were in the room during testing. The experimenter was in the opposite corner of the room sitting on a chair looking at the wall and did not have eye contact with the dog or the handler to not provide any cues to dogs or handlers. During a trial, the experimenter did not make any sound or movement. The experimenter indicated to the handler when a trial started and subsequently the handler cued the dog to search the apparatus. The dog had 99 s to search the apparatus and make a response. Within this time, the handler said aloud the number of the olfactometer port that the dog was alerting to or if the dog did not alert (all clear). The experimenter then indicated with a "yes" if the response was correct and the handler reinforced the dog with their preferred reinforcer. If the response was incorrect,

the experimenter indicated to the handler that the response was incorrect with a "no" and said to the handler which port contained the target. If the handler wanted, the dog was allowed to recheck the olfactometer. During this time the handler was able to immediately mark the correct response. This was done to continue training the dogs within a session and maintain dogs' motivation during testing. The experimenter manually registered the first response reported by the handler into the computer program and the trial was terminated. The testing session was also video recorded to determine if a dog searched a port or not before the handler indicated a response. If the observer determined that the dog accurately sampled a port before a response was scored, we utilized the IR readings to determine the nose poke duration or the amount of time a dog sampled a port. If the experimenter did not observe that a dog sampled a port before the handler called a response, the sniff time was set as zero, because any IR readings recorded were due to interferences. In trials where the handler obstructed the camera view and it was not possible to observe if a dog sampled a port, the IR reading was used because it was the best estimation in those circumstances.

**Positive control test.** After testing with the AHSO, we conducted a positive control test with all dogs. This consisted of testing dogs in a barrel search. The barrel search served as a positive control because all handlers indicated that dogs were proficient in a barrel search. Thus, comparing performance in the olfactometer to the performance during a barrel search will allow us to measure how dogs transferred their training to the AHSO, which was a completely novel task up until that day. The barrel search consisted of placing four 55G barrels in an open room with a 4cm circular hole to allow odor permeation (Fig 2). Three of the barrels were empty and a human volunteer was hidden inside the remaining barrel. The volunteer was instructed to remain as still and quiet as possible to prevent dogs from using auditory or motion cues to identify the correct barrel. The dog team was outside of the room while the experimenters prepared the room. Once everything was arranged the dog team entered the

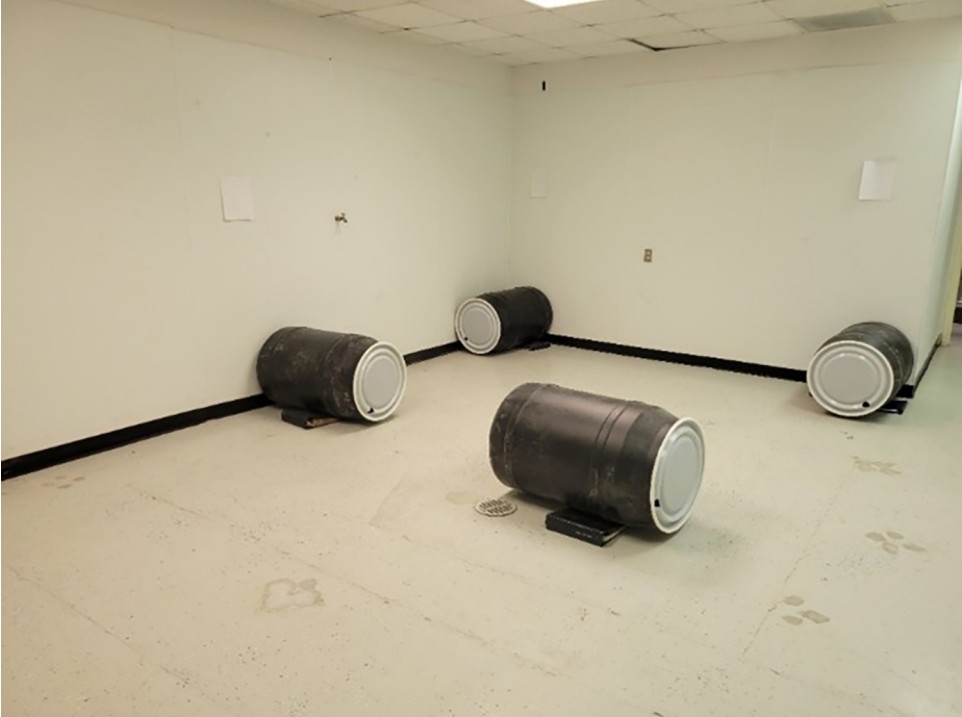

**Fig 2. Positive control test barrel search setting.** Four barrels were placed in an open room. A 4 cm hole was drilled on each lid. Binders on the side of the barrels prevented the barrels from rolling.

room, and the dog performed an off-leash search of the room. The experimenter remained in the room and avoided any interaction with the dog or the handler (Identical to the AHSO). As in the olfactometer, the dog had 99 s to make an alert. If the dog failed to alert within the 99 s the trial was scored as a false all-clear or an odor miss. If the dog alerted to any of the barrels within the allotted time, the handler said aloud the number of the barrel that the dog was alerting to, and the experimenter indicated if the alert was correct or incorrect. Correct alerts were reinforced by the handler and incorrect responses resulted in the termination of the trial. Dogs performed five barrel-searches as part of their control test. During each search, the position of the barrel containing the volunteer was randomized. No blank trials were performed on the barrel searches. The control barrel search was also video recorded. In this test, the observer recorded if a barrel was sampled by the dog. A barrel was considered to be accurately sampled by the dogs if it was observed that the dog introduced its nose into the lid hole in the barrel.

**Variables measured.**   Correct responses, false alerts, false all clears, and dogs' sensitivity and specificity within the AHSO and positive control session were calculated as described in Experiment 1. Herein we utilized the video coding data instead of the IR readings to determine if a dog sampled a port and the number of distractor odors a dog sampled within a trial to calculate specificity and sensitivity.

Both observers watched the video at normal speed and recorded in an Excel sheet their observations. To measure interobserver agreement we calculated the intra class correlation coefficient (ICC). Two blind observers coded the last five trials of the training session in the olfactometer (33% of the total videos in the olfactometer) and the five trials in the positive control. There was substantial agreement between both observers when evaluating if a dog sampled a port, with an ICC of 0.98 for AHSO trials and 0.99 in the control barrel search.

**Statistical analyses.**   A generalized linear mixed model with a binomial distribution was used to evaluate differences in performance between testing methods (AHSO vs Positive Control test). The model predicted the probability of a performance variable (i.e., correct response, false alert, false all clear, sensitivity, and specificity) with the fixed effect of the testing paradigm (AHSO vs control barrel search) and the random effect of dog team.

A liner mixed model was used to evaluate differences in the amount of time dogs sniffed the distractors and the target odor. A visual examination of the data indicated that the raw sniff time measured by the IR sensor to the different odors was heteroscedastic and not normally distributed. Thus, the sniff time recorded by the IR sensors to the different odors were averaged by dog and subsequently $Log_{10}$ transformed. We averaged the sniff time by dog in addition to the log transformation because this was the only model that met parametric assumptions. Only sniff times of ports sampled by the dog within a trial (determined by video coding) were included in the analysis. The statistical model included the fixed effect of odor (HS vs distractors) and the random effect of dog team. A statistically significant difference was declared at $p < 0.05$. If a statistical significance was noted, Dunnett's multiple comparison test was used to test if human scent chamber was statistically different from the sniff time of the distractors.

All statistical analysis were conducted using PROC GLIMMIX in SAS 9.4 software (SAS version 9.4; SAS Inst., Inc., Cary, NC, USA). The reported values in the tables and graphs are the LSmeans, and confidence interval (CL) obtained from the models. For the sniff time analysis, the reported values are the back transformed Lsmeans and CL from the model.

## Experiment 3: Understanding detection dog perception of Human Scent

Experiments 1 and 2 validated the AHSO computer-controlled presentation of human scent to canines. Experiment 3 then used a spontaneous generalization paradigm to evaluate canine generalization from human scent to various components of human scent.

**Participants.** Six dogs that failed to complete a government detection dog program were the participants of this experiment. Three dogs were German shorthaired pointers (2 neutered males and 1 spayed female), and the remaining three dogs were black Labrador Retrievers (3 neutered males). All dogs were naïve to human scent detection but were proficient on the olfactometer task because they were participants of a previous study in our lab.

**Training.** Dogs were trained to operate the AHSO following a similar procedure as Experiment 1. The apparatus presented dogs with a 42.8% air dilution of the headspace of the chamber with the volunteer (i.e., a flow of 1.5 L/m of the chamber headspace was mixed with a 2L/m stream of clean air). Dogs received one or two training or testing sessions in a day depending on volunteers' availability. A training or testing session consisted of 20 trials as the previous experiments.

During a trial containing HS, dogs were trained to alert to the port containing the target odor by holding their nose for 3 s in the port. During blank trials dogs were trained to search all ports, not alert to any, and return to the experimenter.

*Phase 1*: *Initial Training.* Different from Experiment 1, dogs were initially trained to detect and discriminate the headspace of the chamber containing the volunteer from the two distractor chambers. Training under this phase consisted of 20-trials sessions at 100% target odor frequency and 100% reinforcement rate. Dogs continued training until their nose hold was increased from 1 s to 3 s. A total of 5 different volunteers (3 adults male and 2 adult females; 20–23 years) were used in these first training phases. All dogs were able to increase the nose hold to 3s within seven training sessions and all dogs showed an accuracy > 80% at the end of the seventh session.

*Phase 2*: *Novel Targets and Distractor Training.* During the second training phase we introduced cotton gauzes (Dukal Corporation), vinyl gloves, a blank vial, and at least one personal hygiene product that the volunteer within the chamber regularly used (e.g., deodorant, lotion, perfume, shampoo, etc.) as distractor odors. We also decreased the target odor frequency to 80% (i.e., during 4 out of the 20 trials only distractor odors were presented to the dogs). Training during this phase was conducted double-blind. Incorrect response resulted in the termination of the trial without reinforcement. Dogs were trained to return and sit by the experimenter during blank trials. All correct responses were reinforced during this phase. Dogs received five 20-trial sessions in this training phase. Three adult female volunteers (20–23 years) were used for these five sessions. Two were novel volunteers that did not participate during Phase 1 training, and one was a volunteer that participated in the previous training phase. Hence by the end of this phase, dogs were trained with seven different volunteers. All dogs showed accuracy> 80% by the end of the fifth session.

*Phase 3*: *Intermittent Schedule Training.* The third and last training phase consisted of gradually introducing probe trials (intermittent reinforcement) within a session. Probe trials were trials that were preprogrammed to terminate without providing any feedback to the dog. For instance, during a probe trial, independent of the dog's response, an incorrect or correct marker tone was not played. A response was followed just by the trial termination tone and the absence of reinforcement even when it was a correct response. This was to introduce dogs to an intermittent schedule of reinforcement for correct responses and also to familiarize them with the lack of feedback they will encounter during subsequent testing. This was critical to train to allow for the latter assessment of *spontaneous generalization* to various samples without explicitly reinforcing any response to new targets. Thus, this training introduced non-differentially reinforced trials allowing for strong performance to maintain in the absence of feedback on every trial.

As in the previous phase, a session consisted of 20 trials with a target odor frequency of 80%. During the first training session, we introduced 4 probe trials in the session. Three of the

four probe trials were preprogrammed to be during odor trials and one during one of the four blank trials. The computer program randomized which trial served as probe trials within a session. Dogs continued training with these parameters until their performance in two consecutive sessions was > 80% correct responses. After reaching training criterion, we introduced two more probe trials during odor trials and 1 more during a blank trial. Thus, at the end of the training, within a session, there were 7 probe trials. Five of these were preprogrammed to occur during odor trials and the remaining two during blank trials. This was done to minimize the chances of dogs becoming biased towards one response over the other when encountering novel testing odors within probe trials. Adding probe trials without feedback and reinforcement allowed use to give dogs multiple probe trials with novel odors to test their spontaneous response without incidentally reinforcing or extinguishing responses to the testing odors [24].

The same volunteers used in the first two training phases were used in this phase. During Phase 3 and during testing, volunteers showered at the lab with the same soap (Natural, Clear Olive Oil Soap from Life of the Party). This soap was selected because a previous study showed that it does not contain any human specific volatile compounds [29]. After showering, participants waited for 10 minutes in a room before entering the chamber. Participant also wore clean scrubs provided by the lab. The soap with which the participants showered was also used as a distractor odor during training and testing. The distractor chamber contained a set of clean scrubs (the same worn by the volunteers) together with any entertainment source the volunteer had in the chamber. Testing started once dogs' performance was > 80% correct responses during two consecutive sessions with seven probe trials.

*Testing.* Once dogs reached training criterion, we started testing to evaluate dogs' spontaneous response to 14 different testing odors. Table 2 gives a description of the different testing odors.

These were articles scented by the volunteer within the chamber, individual constituents of human scent, and their relevant controls. Articles were impregnated with human scent from the same volunteer in the chamber. This was to ensure that lack of response to the testing odor was not due to discrimination between volunteers. For logistical reasons this was not possible for breath, and for the positive control. Dogs' response to each testing odor was tested in two sessions. A testing session consisted of 20 trials with a target odor frequency of 80%. Additionally, testing sessions were preprogrammed to have 8 probe trials. The odor being tested within a session was presented to the dogs in four of these probe trials. Three of the remaining four probe trials contained the target odor and the remaining probe trial was during a blank trial. Independent of the odor presented during a probe trial and of dogs' response, a probe trial terminated without providing feedback or reinforcement. HS and blank probe trials were also included to maintain responding to the target under an intermittent schedule of reinforcement.

The scentless soap with which the volunteer showered, volunteers 'personal hygiene products, nitrile gloves, Methyl benzoate (CAS # 93-58-3; $10^{-3}$ v/v dilution in mineral oil), and the two distractor chambers were used as distractor odors during testing. The order that the testing odors were tested was randomized but due to logistics, the testing order was the same for all dogs. Dogs received a total of 28 testing sessions in this Experiment (two sessions per testing odor) and a total of 8 probe trials for each testing odor (four probes per session). Dogs received one or two testing sessions within a day depending on volunteers' availability. A total of four volunteers were used during the testing phase, but only one volunteer per session. These volunteers also participated during training. Thus, dogs were already trained with all the volunteers before testing.

*Control tests.* Dogs received a series of positive, vehicle, and negative control tests. The positive control consisted of testing dogs' response to a novel volunteer. The purpose of the

**Table 2. Description of the testing odors.**

| Testing odor | Odor description | Purpose |
|---|---|---|
| Blank gauze | One cotton[a] and one rayon/polyester[b] pretreated[c] gauze (5 x 5 cm) were placed in an odor jar | Control test for all the different tests where gauzes were used as the absorbent material |
| Direct extraction | Volunteer showered with olive oil soap[d]. Ten minutes after showering, the volunteer held three pretreated cotton and three rayon/polyester pretreated gauzes under the arm in the axillar area for 10 minutes. One of each gauze was later transferred into a vial for testing. | Evaluate if an axillary extraction was representative of whole human scent |
| Dynamic extraction | Volunteer showered with olive oil soap. Ten minutes after showering the volunteer entered the chamber. An air pump carried the headspace of the chamber (4 L/m) through a mason jar (1L) containing 15 pretreated cotton gauzes for 2 h. Five gauzes were placed in an odor jar for testing. | Evaluate if cotton gauzes could be impregnated with the headspace of the chamber and used as training aids. |
| 2h-Direct extraction | Volunteer showered with an olive oil soap and did not wear any personal hygiene product overnight. In the morning, the volunteer taped three pretreated cotton and three pretreated rayon/polyester gauzes under the arm for 2h. One of each gauze were later transferred in a vial for testing. | Evaluate if increasing the extraction time will result in a better representation of whole human scent |
| Positive control | Two novel volunteers showered with olive oil soap. One volunteer was assigned to be in the target chamber and the other volunteer was in the probe chamber. During probe trials a pump carried out the headspace of the probe chamber and presented it to the dogs in the other room. | Evaluate generalization to a different and novel volunteer within a session. This was to ensure dogs were able to generalize to novel whole human scent. This test allowed us to confirm that generalization was achievable with the paradigm if the testing odor resembles whole human scent. |
| CSSB (Cotton-Shirt-Socks-Breath) | Volunteer showered with an olive oil soap and did not wear any personal hygiene product. Volunteer taped 3 pretreated cotton gauzes under their arm and wore a scrub shirt[e] and clean socks for 18–24 h. Before testing a piece of the scrub shirt and socks (5 x 5 cm), and one gauze were placed and sealed in an odor vial. In addition, the volunteer blew into the jar 5 times for 10 s. The vial was then used for testing. | Evaluate if different fabrics exposed to different parts of the body together with breath were representative of whole human scent. |
| Control CSSB | A piece (5 x 5 cm) of clean sock, scrub shirt, and pretreated cotton gauze were placed in an odor jar. | Control test for CSSB. Any difference in response between the two tests will be due to Human scent captured by the fabric. |
| Breath | A volunteer rinsed its mouth with RO water five times for 30 s to remove any food or drink odor. During testing the volunteer exhaled into a jar (500 mL) and a pump carried the headspace of the jar containing the exhaled breath to the odor ports in the other room during a probe trial. | Evaluate if breath was representative of whole human scent |
| Breath Control | A jar was filled with 100 ml of RO water. Filtered air (1L/m) was inserted to the bottom of the jar to produced bubbles and a pump carried the humidified air to the port in the other room during probe trials. | Control for breath test. Ensure dogs were responding to odor and not to potential differences in humidity. |

*(Continued)*

**Table 2.** (Continued)

| Testing odor | Odor description | Purpose |
|---|---|---|
| GETXENT | Three GETXENT tubes were impregnated with human scent by having them in the chamber with a volunteer for 2h. | Evaluate if GETXENT tubes can capture and release volatiles representative of whole human scent |
| Control GETXENT | Blank GETXENT tubes. | Control for GETXENT test |
| Hair | A volunteer cut its hair and immediately 0.5 g of hair were placed in an odor jar and use for testing. | Evaluate if hair was representative of whole human scent |
| SOC (Scent of Chamber) | A volunteer wore a shirt[f] inside the chamber for 2h. The shirt was cut in three pieces and each piece placed on a jar (1L) for testing. | Evaluate if a shirt was able to capture volatiles representative of whole human scent |
| Control SOC (Scent of Chamber) | A clean shirt was placed for 2h in the chamber with no volunteer in it. | Control for SOC test |

[a]Dukal Coorporation.

[b]Band-Aid, Johnson&Johnson.

[c]Spiked with methanol solvent and placed in the oven at 105 C˚ for at least 2h to remove odor contamination.

[d] Natural, Clear Olive Oil Soap from Life of the Party.

[e]Scrubstar®.

[f]Hanes, 60% Rayon and 40% polyester.

positive control was to ensure that dogs were able to generalize to novel whole human scent within a session. This test allowed us to confirm that spontaneous responses were achievable with the paradigm if the testing odor satisfactorily resembled whole human scent. Vehicle controls tested dogs' response to the different absorbent materials ("vehicles" for odor) itself. The desired outcome of these control tests was a lack of response to the all the absorbent materials tested and a robust response to the positive control. A lack of response to the vehicles/absorbent materials indicates that responses to the testing odors were due to the presence of human specific volatiles captured during the impregnation procedure and not due to the absorbent/ vehicle materials. Strong response to the positive control test indicates that with the method and apparatus used, we were able to produce strong responses if the testing odor was perceptually similar to the headspace of the chamber with the volunteer. Thus, strong response to the positive control and not to a test odor would indicate that the testing odors were not representative of whole human scent. In addition, we conducted a 10-trial negative control test with no volunteer in the chamber at the end of the experiment as described in Experiment 1 to confirm dogs were not using unintentional cues.

**Statistical analysis.** Within a testing session we measured the probability of a dog alerting to the target odor (volunteer in the chamber) and to the testing odor during the 4 probe trials. The probability of alert was calculated by dividing the number times a dog alerted to an odor over the total number of trials where the odor was present. Blank trials were not included in the statistical analysis because the main goal of the experiment was to evaluate and compare dogs' response rate to the testing and target odors. Using the IR sensors in front of the ports, we also calculated the amount of time a dog spent sniffing the port containing the target and the distractor odors as described in previous experiments. In addition, we evaluated the probability of false alerts and false all clear within a session.

Due to the low responses to the different testing odors a generalized linear mixed model with a binomial distribution was not able to fit the probability of alert data because the model was not able to converge. Thus, to evaluate the effect of testing odor on the probability of alert, we averaged dogs' responses to the testing and target odors within each session. We then fit a linear mixed model to evaluate the effect of testing odor and testing session. The model included the fixed effect of testing odor, testing session (i.e., first or second testing session), and their interaction. Dog ID was included as a random effect in the model. Tukey's multiple comparison test was used to evaluate statistical difference in dogs' sniff time and alert rate to the different target odors. A statistical significance was declared at $p < 0.05$.

We averaged the amount of time the dogs spent sniffing each testing odor within a session. Different from the previous experiment, a log transformation was not applied because it did not improve the normality or homoscedasticity of the averaged data. The same linear mixed models described above was used to evaluate the effect of testing odor on the sniff time measured by the IR sensor.

A binomial test was conducted to evaluate if the probability of alert to the testing odors was above chance levels ($H_0$ $p = 0.33$). The Conchran's Q test was used to evaluate the effect of multiple unreinforced presentations of the testing odors on dogs' response rate. The model included the fixed effect of probe trial number (1–8) and dog ID as a block effect. This model was run independently for each testing odor. This analysis was to evaluate if extinction of dogs' responses occurred after multiple unreinforced trials. The linear mixed model and the binomial test were conducted using PROC GLIMMIX and PROC FREQ in SAS 9.4 software (SAS version 9.4; SAS Inst., Inc., Cary, NC, USA), respectively. The Conchran's Q test was conducted using the non.par library in R.

## Experiment 4: Further evaluation of volatiles of breath and skin origin as main constituent of human scent

Results from Experiment 3 suggested that breath is an important constituent of human scent (see Results section). The aim of Experiment 4 was to further evaluate dogs' spontaneous responses to breath and to a volunteer in the chamber when breath was removed from the headspace of the chamber via a snorkel. This was an attempt to further dissect and understand the value of volatiles of breath and of skin origin as constituents of human scent. The method utilized in Experiment 4 was further modified to minimize the chance of within session contrast between the target and the testing odor while preserving a spontaneous generalization paradigm.

**General methods.**   The same dogs that participated in Experiment 3 participated in this experiment. We also used the same apparatus and settings as described above. For this experiment we only utilized two volunteers (Adult Females 20–23 years) as the target and testing odors. The volunteers also participated in the previous experiments; thus, dogs were already familiarized with the volunteers and their detection performance for both volunteers was > 90% correct responses. Celery powder, and petroleum jelly were used as novel distractors during this experiment. Dogs received up to 4 training or testing sessions within a day.

**Training.**   Training sessions consisted of 10 trials. During training the participant was in the chamber as described above. The number of blank and probe (non-reinforced) trials within the 10-trials session varied from 0 to 4. This was to reduce the probability of dogs discriminating between a testing (non-reinforced) and training (reinforced) session based on the number of probe and blank trials. If dogs' performance during a training session was > 0.80 correct responses, they moved to testing, otherwise training repeated until meeting criterion. Dogs received one training session between each testing session to maintain motivation and

performance in subsequent testing sessions. The goal of these training sessions was to provide some reinforced trials to the dogs between testing to maintain robust responding. The goal of this procedure was to provide as many non-reinforced probe trials to assess spontaneous generalization to the testing material, while maintaining robust response to the trained targets.

**Testing.** Testing consisted of 5 non-reinforced trials where the testing odor was presented in one of the three olfactometer ports and the other two ports contained a distractor odor. Independent of dogs' responses, all testing trials terminated with an end of trial tone without correct or incorrect feedback and reinforcer. This was done to ensure we were evaluating spontaneous generalization from the whole human scent target, and not simply reinforcing dogs to respond to a novel odor variation.

We tested dogs' response to three testing odors. Fig 3 shows a schematic of the testing procedure. Each odor was tested twice, each time with a different volunteer. Thus, in total dogs received two testing sessions for each testing odor (10 probe trials in total).

After meeting training criterion and before testing, dogs received a positive control test with each volunteer. This consisted of 5 consecutive unreinforced probe trials where the target odor was the volunteer in the chamber as in training. This positive control (Positive control 1) was to evaluate if dogs were able to maintain responses during 5 consecutive unreinforced trials. This was to ensure that the new paradigm was able to produce robust responses if the target odor was perceptually similar to the headspace of the chamber. Dogs had to correctly alert to four out of the five positive control probe trials before testing with the other testing odors. Because testing sessions were run under extinction conditions, dogs had to score > 0.80 correct responses in a subsequent training session before receiving the next test. Dogs continued

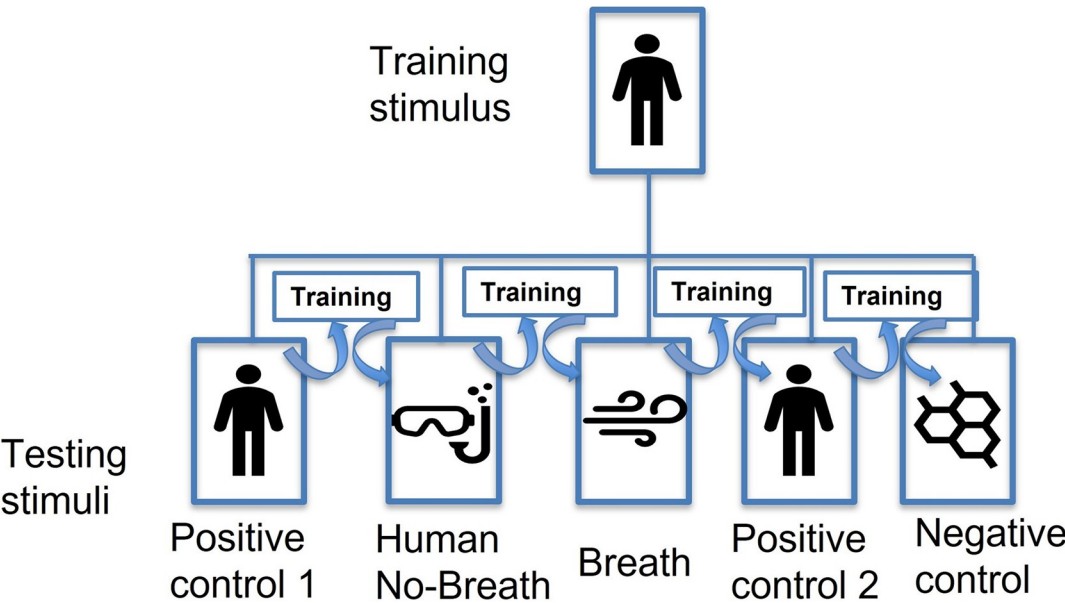

**Fig 3. Schematic of the training and testing procedure.** During testing one of two volunteers were in the chamber and dogs were trained to whole human scent. The same two volunteers served as the source for the testing stimuli. During both positive controls dogs received 5 testing trials with the volunteer in the chamber as in training. During the Human No-Breath test the volunteer was in the chamber with a snorkel and a scuba mask. The breath of the volunteer was exhausted from the chamber through a PVC pipe attached to the side of the chamber during testing. A pump carried the headspace of the volunteer without breath to an olfactometer port. During the breath test a volunteer exhaled in a vial and a vacuum pump carried the breath to the olfactometer port. The distractor control consisted of presenting dogs with a novel random odor (Isobutyl propionate). Dogs received a training session in between testing sessions to ensure they still were responsive to the whole human scent target.

in training until they met training criterion to ensure performance and search behavior under the superimposed intermittent schedule of reinforcement. After meeting training criterion, dogs received two testing sessions where we tested their response to only breath and to a participant in the chamber with no breath (Human No-Breath). The order of testing was alternated between volunteers.

To remove the breath from the chamber the volunteer wore a scuba mask and a snorkel within the chamber. The exhaled air was then carried outside the chamber through a PVC pipe on the side of the chamber that was connected to the end of the snorkel. A scuba mask and a snorkel were added inside one of the distractor chambers to be used as distractors. The scuba mask was used to prevent accidental exhalation through the nose. The same method in Experiment 3 was used to test breath.

**Positive and distractor control test.**   After the completion of Experiment 4 we conducted additional control tests. The distractor control test consisted of testing dogs' response to a novel odor in five unreinforced probe trials. We utilized Isobutyl propionate (diluted in mineral oil $10^{-3}$; CAS # 540-42-1) as a novel odor. This odor was selected based on availability. The distractor control test was to ensure dogs were not alerting to a novel odor that could just be perceived different from the distractor odors. If dogs' responses to the testing odors was dependent on perceptual similarity to human scent, we were expecting them to show no responses to the distractor control test.

A final positive control (Positive control 2) was conducted as described above. This test was to ensure that after multiple testing sessions under extinction, dogs were still responsive to the target odor. Similar response rates during the first and second positive control test will indicate that the procedure did not produce extinction of search behavior or responses. This would further confirm that difference in response rate to the testing odors were due to perceptual differences and not due to extinction of responses induced by the paradigm.

**Statistical analyses.**   As in Experiment 3, a generalized linear mixed model with a binomial distribution did not converge when we tried to fit the probability of alert data. For this reason, we averaged each dog's probability of an alert to the testing and control odors during each testing session. A preliminary linear mixed model was used then to evaluate the effect of testing odor, testing session, and their interaction on the probability of alert. This preliminary model showed that the effect of session and the interaction between session and testing odor were not statistically significant. Hence, we removed both factors from the final statistical model. The final linear mixed model included the fixed effect of testing odor and dog ID as a random effect. The same model utilized for the probability of alert was used to evaluate the averaged sniff time.

## Results

### Experiment 1

Fig 4. shows the average training progression of the two participants, Buster and Stella, in the automated human scent olfactometer (AHSO) developed. Both dogs successfully completed all training and testing phases within 28 sessions (~ 13 days). By the end of the second session of Phase 1, the nose hold criterion was increased to 3 s for both dogs. This suggested that learning of the headspace of the human chamber occurred quickly. The average proportion of correct responses, false alerts, and false all clear during Phase 1 were 0.93 ± 0.018 (SEM), 0.05 ± 0.016 (SEM), and 0.02 ± 0.009 (SEM), respectively.

During the first two sessions of Phase 2, the proportion of trials ending in a false alert increased from 0.075 during the last session of Phase 1 to 0.15 ± 0.05 (SEM). This was a result of adding novel distractors. When the number of false alerts were subset by distractor odors,

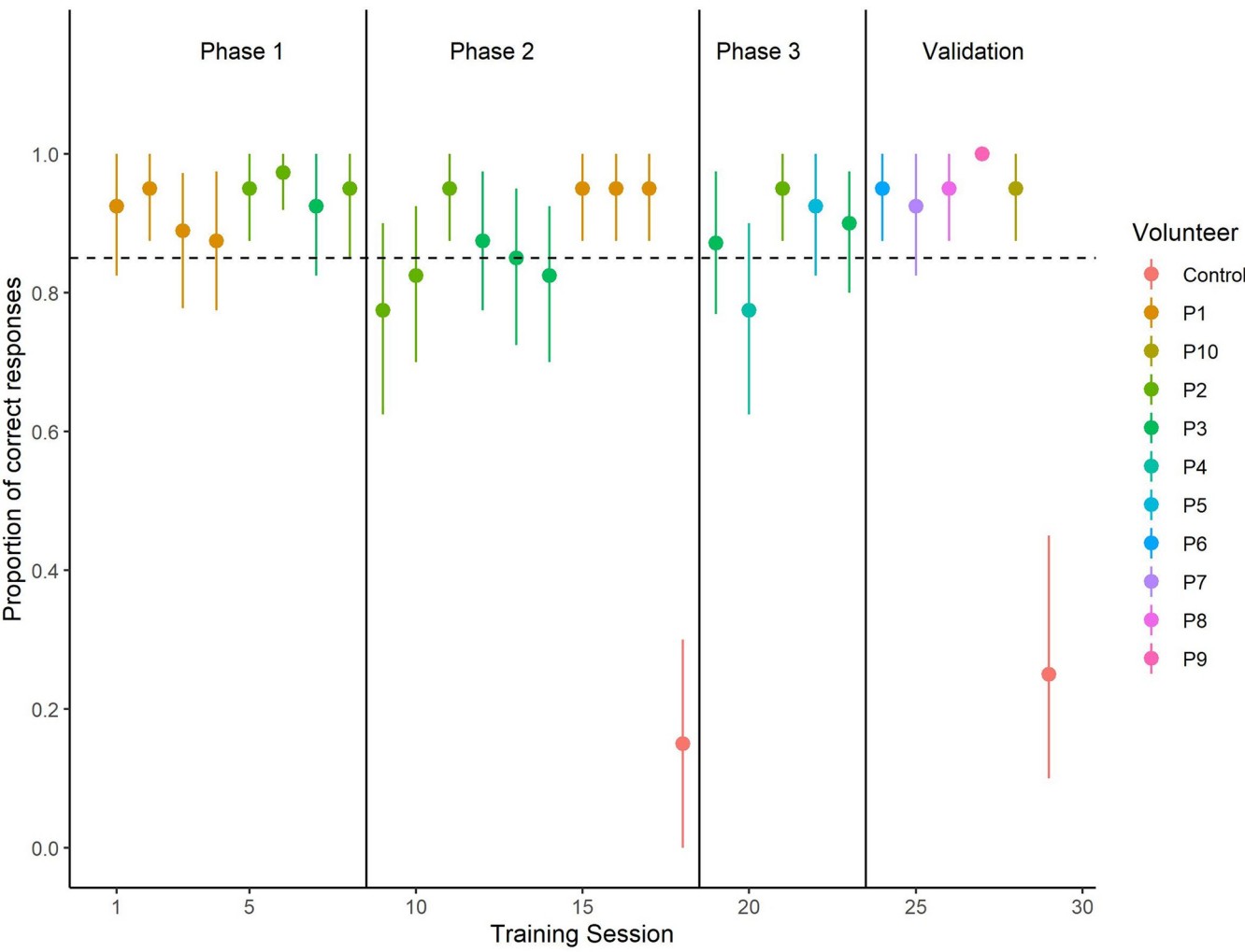

**Fig 4. Dogs training progression.** Vertical lines indicate the initiation of the different training Phases. The color of the data point indicates the volunteer that served as the human scent (HS) source. Dashed line indicates 0.85 proportion of correct responses, our training criterion. Dogs completed their training within 28 sessions. At the end of the experiment, the proportion of correct responses for both dogs was> 0.90.

we noticed that 70% of them were to the personal hygiene products (PH) of the volunteers and the remaining false alerts were mainly to the distractor and blank chambers (Fig 5B). After the first two sessions of Phase 2, the false alert rate decreased, and no false alerts occurred during the last three sessions of this phase (Fig 5A). Overall, false alerts to the PH and the distractor chambers represented more than 90% of all the false alerts during this phase. During the last session of this phase, both dogs scored 95% correct responses, zero false alerts, and 5% false all clears.

**Experiment 1: Negative control test.** Dogs made almost entirely all clear responses during the negative control test (19 out of 20 trials), where no volunteer was in the chamber. This yielded an accuracy of $0.15 \pm 0.082$ (SEM), reflecting dogs made correct "all clear" responses during the two blank trials of the session. Most importantly, no responses were made to the port containing the headspace of the human chamber when no volunteer was present. This indicates that dogs' performance was highly dependent on the presence of a human in the chamber and not on unintentional cues from the olfactometers such as sounds, pressure, or air flow differences.

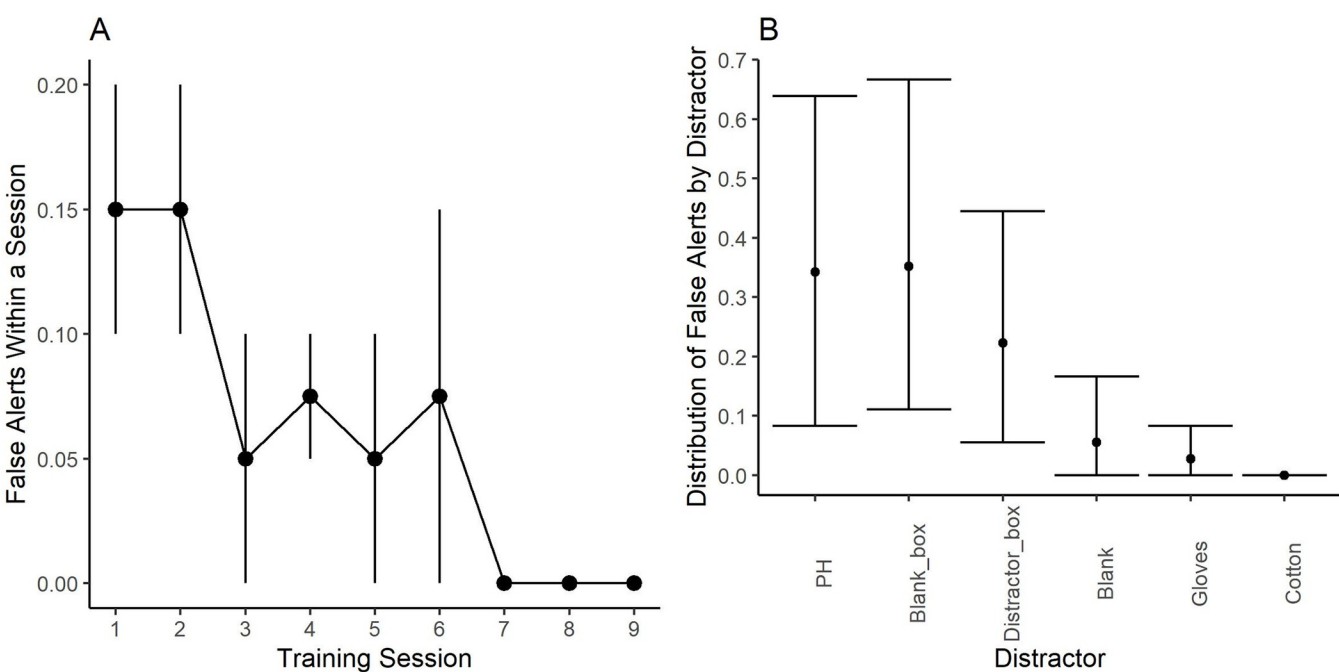

**Fig 5. False alerts during Phase 2.** Data points are the mean ± 95% confidence intervals. Introducing novel distractor odors increased the proportion of false alerts during the first two sessions. After the second session with novel distractors the proportion of false alerts reduced and by the end of the phase dogs did not show any false alerts to the distractors. The distribution of false alerts by distractor odor shows more responses to the person hygiene (PH) products of the volunteers and to the control chamber than to the other distractors.

In Phase 3, training under an intermittent schedule of reinforcement did not produce a significant change in the overall performance relative to the performance observed in Phase 2. Further, introducing limonene and isoamyl acetate as novel distractors during Phase 3 did not increase the number of false alerts. Both dogs transitioned to the intermittent schedule of reinforcement without any decrement on performance. Buster transferred to the two novel volunteers in this phase without showing a significant performance decrement (proportion of correct responses > 0.80). Stella showed a decrement in performance (0.70 correct responses) when tested with Volunteer P4. This was due to a high false alert rate to one of the PH products of this volunteer. However, this decrement in performance was not observed when tested with volunteer P5. During the last session of Phase 3, the average proportion of correct responses, false alerts, and false all clears were 0.90 ± 0.10 (SEM), 0.075 ± 0.075 (SEM), and 0.025 ± 0.025 (SEM), respectively.

Stella's and Buster's proportion of correct responses was ≥ 0.90 during all the sessions in the validation test using novel volunteers. No performance decrement was observed as a result of testing dogs with novel volunteers. This indicates that after previous training with only five volunteers, both dogs were able to successfully generalize to the scent of five novel volunteers. The overall proportion of correct responses, false alerts, and false all clears were 0.95 ± 0.14 (SEM), 0.015 ± 0.007 (SEM), and 0.030 ± 0.011 (SEM), respectively. Overall sensitivity and specificity to HS during the testing phase were 96.25% (95% CI- 92.02% - 98.61%) and 98.97% (95% CI- 97.79%- 100%), respectively.

The results of the second negative control test again indicated that when no human was present in the target human scent chamber, dogs failed to identify the headspace of that chamber in 15 out of the 16 trials where it was presented. The only responses scored correctly were all clear responses on blank trials and one correct response to the port containing the headspace of the empty target chamber.

**Table 3. Performance of SAR teams (N = 5) in the AHSO and the Control test.**

| | AHSO | | Control Barrel search | | *F* | *P* value |
|---|---|---|---|---|---|---|
| | LSmean | CL | LSmean | CL | | |
| Correct responses | 0.82 | 0.68–0.90 | 0.92 | 0.72–0.98 | 1.26 | 0.26 |
| False alerts | 0.09 | 0.02–0.31 | 0.03 | 0.002–0.25 | 1.22 | 0.27 |
| False All clear | 0.075 | 0.02–0.21 | 0.04 | 0.005–0.24 | 0.32 | 0.57 |
| Sensitivity | 0.97 | 0.81–0.99 | 1 | * | 0.00 | 0.99 |
| Specificity | 0.96 | 0.20–0.99 | 0.99 | 0.31–1 | 1.76 | 0.27 |
| Correct responses during odor trials | 0.85 | 0.68–0.94 | 0.92 | 0.72–0.98 | 0.68 | 0.41 |
| Correct responses during blank trials | 0.70 | 0.35–0.93 | . | . | | . |

CL = 95% confidence limits.

*No variability to calculate confidence intervals.

. No blank trials were given to dogs during the control barrel search.

**Experiment 2: Search and rescue canine validation.** Five of the eight participant teams successfully learned to operate the AHSO after only 15 training trials. Two of the three teams that did not meet training criterion were voluntarily removed from the experiment by their handler because they considered that their dogs did not learn the task within the allotted time. The third team was considered to not have acquired the task based on the video coding data from both observers. This dog only sampled the olfactometer ports (introduced the nose in the port) in one of the five last training trials.

Table 3 shows the average performance in the AHSO and the control barrel search. The main effect of testing method (e.g., AHSO vs barrel search) was not statistically significant (P > 0.05) for any of the performance variables measured, indicating performance was similar in the barrel search and the novel AHSO. The overall proportion of correct responses in the AHSO was 0.82 ± 0.05 (SEM). Although not statistically significant, this was ~10% lower than the proportion of correct responses in the control barrel search. Similarly, the proportion of trials ending in a false alert and false all clear were 6% and 3% higher in the AHSO compared to the control barrel search, respectively. When the data was subset and only the performance during odor trials was evaluated (e.g., blank trials were not included in the analysis which were not conducted for the barrel test) the proportion of correct responses in the AHSO increased to 0.85.

The effect of testing method on dogs' sensitivity and specificity was also not statistically significant. We found that dogs' sensitivity to HS was 100% in the control barrel search and 97% in the AHSO. This measurement indicates that if a dog sampled the port containing HS (determined by a blind observer), 97% of the time the handler reported the correct response. Likewise, dogs' specificity to HS was 99% in the control barrel search and 96% in the AHSO. This measurement also indicates that if a dog explicitly searched a port with a distractor odor, they rejected it 96% of the time in the AHSO. It is important to note, the barrel search did not contain distractor odors, whereas the AHSO included 5 distractor odors.

Fig 6 shows the overall nose poke duration or the average time dogs spent sniffing a port containing the headspace of the chamber with the volunteer and the different distractors. The main effect of odor was not statistically significant (F = 1.30; P = 0.24). Nevertheless, on

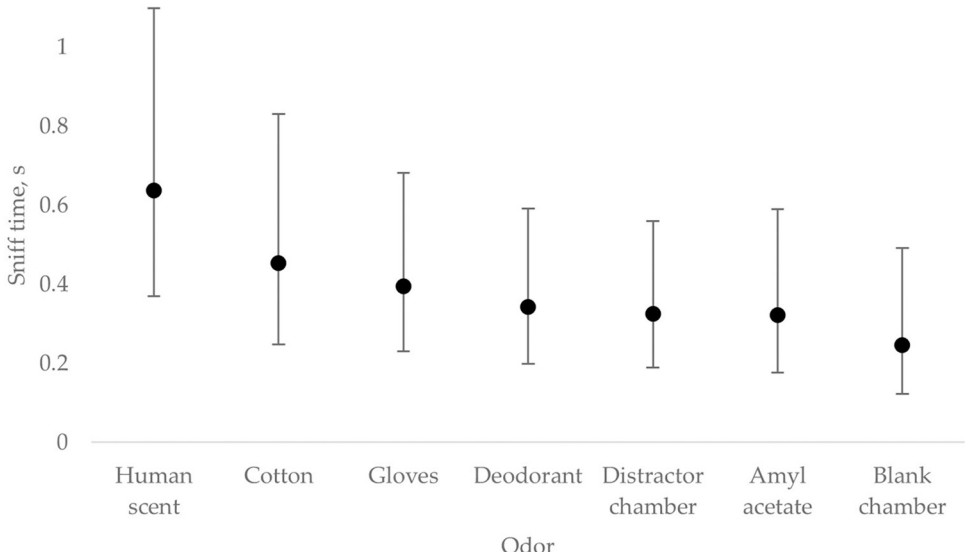

**Fig 6. Back transformed LSmeans ± 95% confidence intervals of the amount of time (s) dogs spent sniffing the target odor (Human scent) and the different distractors during a trial.** The main effect of odor was not statistically significant (P = 0.24). However, on average, dogs sniffed more time to the human scent than all other distractors.

average, dogs sniffed the port containing HS at least 0.20 s longer than any other distractor odor. Furthermore, exploratory analysis of the data showed that on trials where the handler reported a correct response, on average, dogs sniffed the odor port containing HS 0.80 ± 0.10 s. This average sniff time was at least 0.31 s more than to any of the other distractors during correct odor trials. Correspondingly, the video coders did not observe the dog inserting its nose in the port containing the target odor in five of the six incorrect odor trials. This indicates that most errors were due to inaccurate sampling of the ports. This could have been due to the novelty of the task and the limited training time.

**Experiment 3.** All dogs were able to complete the 28 spontaneous generalization testing sessions successfully. Over the 28 sessions, the average alert rate to the trained target odor was 0.97 ± 0.04 (SEM) and the proportion of trials ending in false alert was 0.006 ± 0.001 (SEM). Similarly, the false all clear rate during testing was low (0.024 ± 0.003 (SEM)). This indicated that dogs were under strong stimulus control and were responding to the headspace of the chamber with the volunteer.

The main effect of odor (F = 60.32; Df = 14; p <0.001) and testing session (F = 8.14; Df = 1; p = 0.005) on the probability of alert to the different testing odors were statistically significant but their interaction was not (F = 1.09; Df = 14; p = 0.37). Overall, the probability of an alert to the different testing odors was higher in the first testing session than in the second testing session. The overall probability of alert to the probe samples during the first session was 0.19 ± 0.03 (SEM) *vs* 0.12 ± 0.03 (SEM) in the second session.

Dogs' overall alert rate to the target odor (i.e., the headspace of the chamber with the volunteer; 0.97 ± 0.04 (SEM)) was statistically higher than the probability of alert to any of the testing and control odors (Fig 7). Although dogs alert rate to the positive control test with a novel volunteer was statistically significantly lower than their alert rate to the target odor (0.75 ± 0.04 (SEM)), it was statistically higher than to all other testing odors. Dogs' probability of an alert to all testing odors except breath was < 0.10. The overall probability of an alert to breath was 0.39 ± 0.12 (SEM). This was statistically higher than their responses to the breath control (0.00) but statistically lower than their responses to the target and the positive control. The

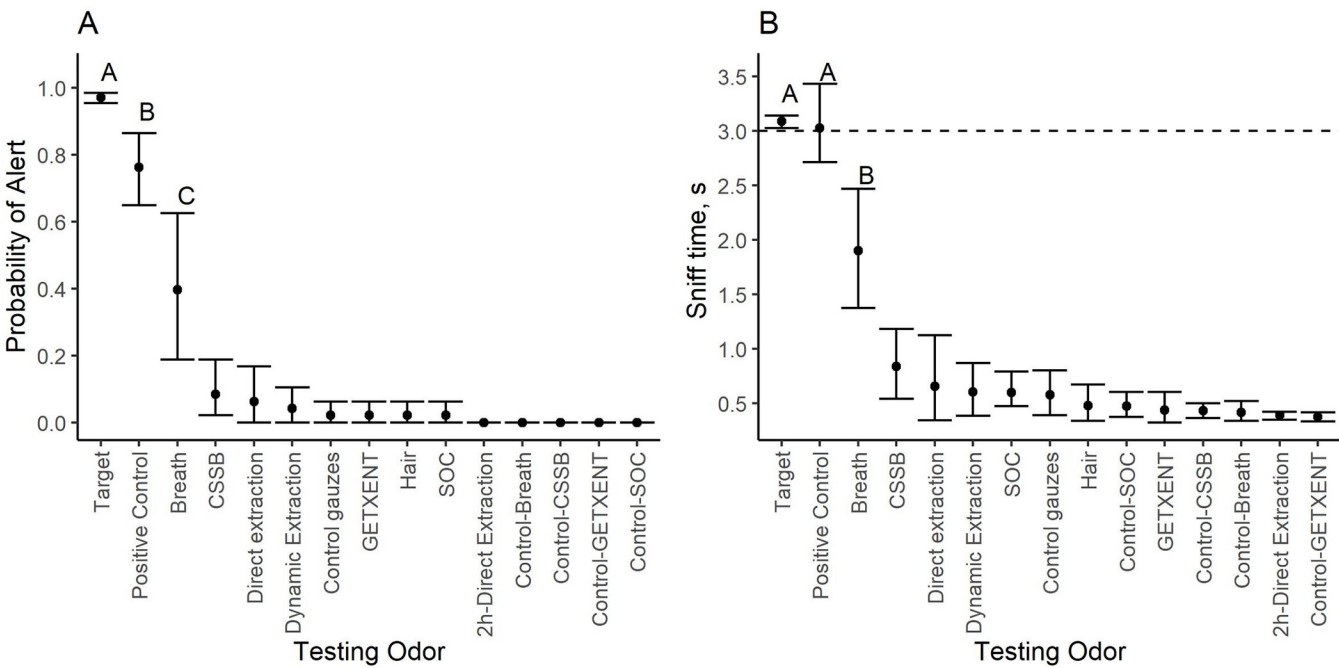

**Fig 7. Dogs (N = 6) average probability of alert and sniff time ± 95% confidence interval to the different testing odors.** Data points with different superscripts within a panel are statistically different from each other. A) shows dogs response rate to the different testing odors. B) Shows dogs sniff time (s) to the target and testing odors. Dashed line indicates the 3 s mark which was our alert criterion.

probability of alert to all other testing odors was not statistically different from their relevant controls and significantly lower than their response to the target odor, positive control and breath.

The effect of testing odor (F = 58.29; Df = 14; p < 0.001) and session (F = 16.49; Df = 1; p < 0.001) were also statistically significant when evaluating the sniff time (Fig 7B). Different from the analysis of the probability of alert, the interaction between testing odor and session showed a tendency towards statistical significance (F = 1.67; Df = 14; p = 0.06). The overall sniff time to the testing odors was higher during the first session (1.08 ± 0.11 s) than during the second one (0.80 ± 0.09 s). Dogs' sniff time to the target odor (3.08 ± 0.03 s) was not statistically different from their sniff time to the positive control (3.02 ± 0.19 s), but both were statistically higher than the sniff time to all other testing odors. Of the remaining testing odors, dogs showed an increased sniff time to breath (1.89 ± 0.30 s) relative to the breath control (0.42 ± 0.05 s) and to all other testing odors. The sniff time of all other test odors did not differ from the sniff time to their relevant controls, and it was less than 1s in duration.

The probability to alert to the testing odors did not differ based on the trial. Although not statistically significant (Q = 7.57; Df = 7; p = 0.37), a visual examination of the data (Fig 8) showed that the probability to alert to breath gradually decreased from 0.87 ± 0.14 (SEM) during the first trial to around 0.21 ± 0.05 (SEM) in the remaining of the trials. A similar reduction in the probability of alert as probe trials progressed was also observed for the positive control where it decreased to 0.50 during the last probe.

A significant drop in performance was observed during the negative control test where we tested dogs with no volunteer in the chamber. Dogs made primarily all clear responses (47% of trials) or was a chance in selection between the three ports (23% correct).

**Experiment 4.**   One of the six dogs that participated in the experiment was removed from the data analysis because she showed extinction of search behavior and did not respond to any

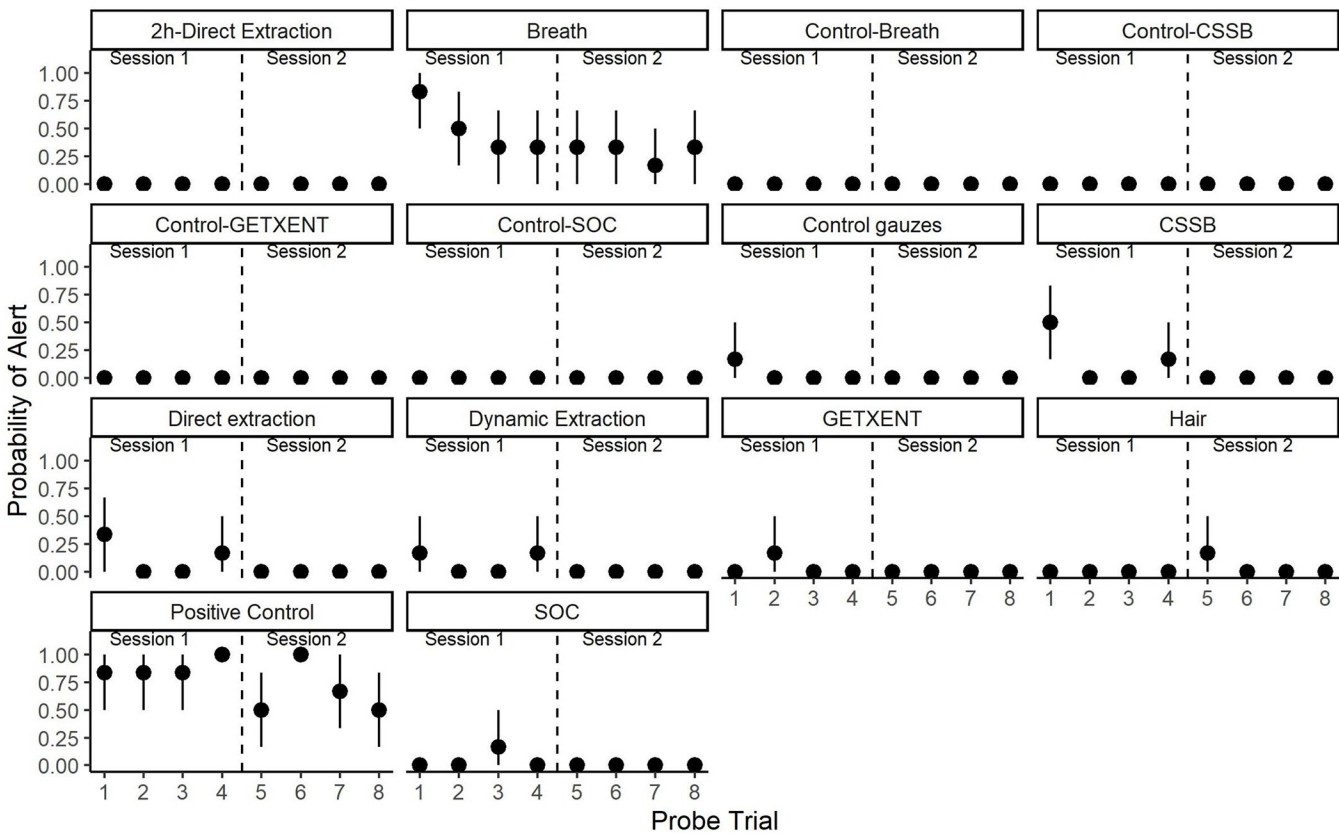

**Fig 8. Averaged probability of alert to the different testing odors during a trial ± 95% confident interval.** Dashed line indicates the different sessions.

of the probe trials during the second positive control test. Her lack of response during the second positive control indicates that the paradigm induced extinction of search behavior and responses. Thus, we removed her from analysis because her lack of response could be a behavioral effect (extinction of the search and alert behavior) rather than an effect of perceptual differences between testing odors. None of the remaining dogs showed any performance decrement during training or the second positive control. During the training sessions in between testing sessions, the performance of the remaining 5 dogs was $> 0.80$ and none of the dogs needed more than one training session in between testing sessions to meet training criterion. Thus, with the exception of the dog that was excluded, we did not observe any negative behavioral or performance decrement due to the paradigm used. The overall proportion of alerts to the trained target was $0.98 \pm 0.004$ (SEM) and false alerts and false all clears during the different training sessions were $0.003 \pm 0.003$ (SEM) and $0.01 \pm 0.006$ (SEM), respectively. During testing sessions, the overall proportion of false alerts to distractor odors was $0.004 \pm 0.003$ (SEM).

The effect of session was not statistically significant. The main effect of testing odor (Fig 9) was highly significant for the probability of alert (F = 35.45; Df = 5; p < 0.001) and sniff time (F = 43.86; Df = 5; p < 0.001). The probability of alert to whole human scent (e.g., volunteer in the chamber with breath) during training ($0.98 \pm 0.004$ (SEM)) was not statistically different than their response rate to whole human scent during the first ($0.96 \pm 0.02$ (SEM)) and second ($1.00 \pm 0.00$ (SEM)) positive control tests and their overall response rate to breath only ($0.88 \pm 0.07$ (SEM)). The probability of alert to whole human scent (during training and

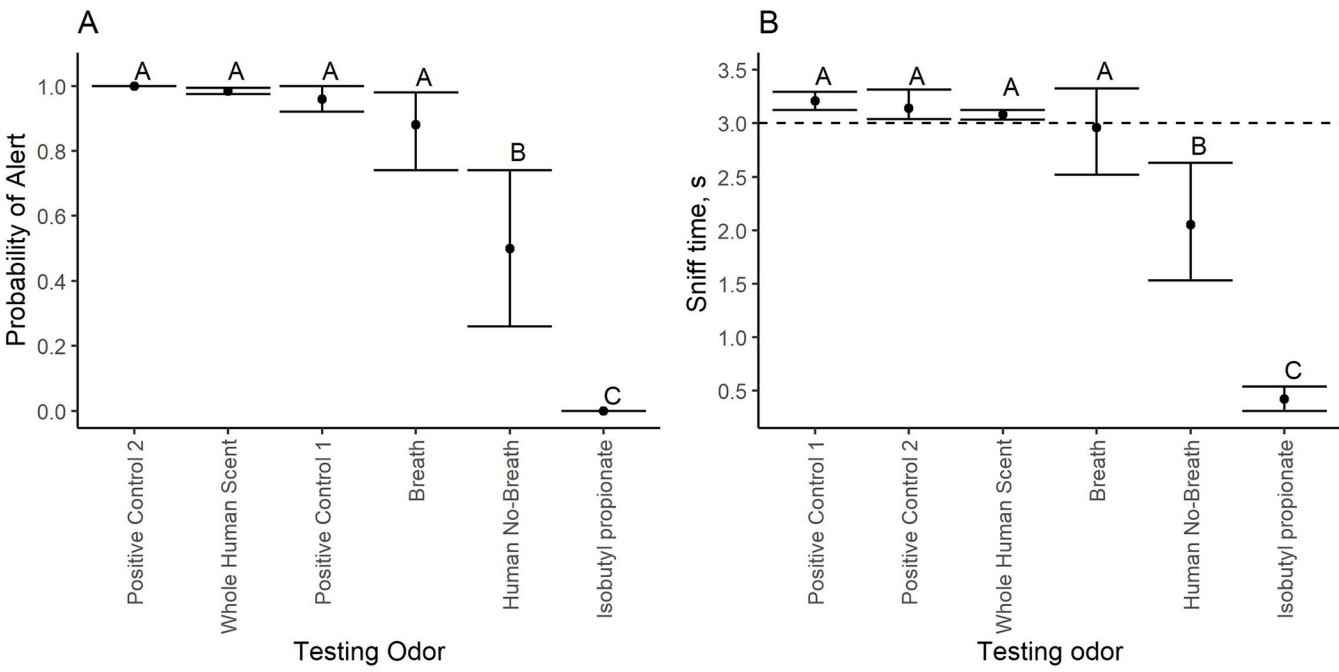

**Fig 9. Dogs (N = 5) averaged probability of alert and sniff time ± 95% confidence interval to the different testing odors and to whole human scent during training session in between testing sessions.** Data points with different superscripts within a panel are statistically different from each other. A. shows dogs' probability of alert to the different testing odors. B) Shows dogs sniff time (s) to the testing odors. Dashed line indicates the 3 s mark which was our alert criterion. Sniff time above 3 seconds indicated that dogs sampled the ports multiple times.

control tests) and to breath were statistically higher than their probability of alert to a volunteer in the chamber when their breath was removed (Human No-Breath; 0.50 ± 0.14 (SEM)) and their probability of alert to the negative control (0.00). The analysis of the sniff time also showed similar results (Fig 9B). Overall, the amount of time dogs spent sniffing whole human scent during training was not statistically different than the sniff time in both positive controls and the amount of time they spent sniffing breath. Dogs spent less time sniffing the olfactometer ports when presented with the headspace of the chamber with the volunteer with no breath (Human No-Breath) relative to breath only, the positive control (Human+Breath), and whole human scent during training. However, the sniff time for the human with no breath was higher than the sniff time for the negative control.

## Discussion

Results from Study 1 show that two naïve dogs were able to learn to detect and discriminate the headspace coming from a chamber with a human volunteer from two identical chambers with no volunteer and other distractor odors with just two weeks of training. Further, control testing showed that dogs did not alert to the human chamber if a volunteer was not in the chamber. This result suggest that the chamber was able to capture human specific volatiles (e.g., sweat, breath, raft, etc.) produced by the volunteer and that the AHSO was able to present the headspace of the chamber containing these volatiles to dogs in the other room and that they readily learned to discriminate the headspace of the chamber containing human specific volatiles from distractor chambers and odors.

The personal hygiene products (PH) of each volunteer were added as distractors during Phase 2. During each session, two PHs were used as distractors. This was to ensure dogs were alerting to human specific volatiles and not to other odors associated with the volunteers.

During the first two sessions where the distractor chambers and the PHs were introduced as distractors, we observed an increment in the number of false alerts. More than 70% of the false alerts observed were to the PH of the volunteer within the chamber. The increase in false alerts to the PH further confirms that dogs were alerting to the scent produced by the volunteer in the chamber. For instance, even when we instructed the volunteers to not wear any PH for at least 4 h before testing, it could be that traces of the PH odor could still be present in the volunteer skin or clothing. Hence it would have not been unlikely that the headspace of the chamber could contain traces of the PH odor. Traces of PH on the volunteer skin or cloth, and thus in the headspace of the chamber, can explain why dogs showed false alerts to them. Despite this, dogs quickly learned to discriminate the odor of the PH from the headspace coming from the chamber with the volunteer. Altogether the data indicate that after the first two sessions of this phase, dogs learned to ignore (not alert) the odor of PH.

After training with just five volunteers, both dogs' performance to five novel volunteers was ≥ 90% under a 65% reinforcement rate (i.e., 35% of correct responses to the novel volunteer were not reinforced in the session). Stella and Buster had an overall sensitivity and specificity ≥ 90% during this phase. This suggests that dogs were able to transfer their prior training to novel volunteers and maintained proficiency. Their false alert rate to the novel PH used in the validation phase was never greater than 2.5% during any of the testing sessions (e.g., 1 false alert in 40 trials). This suggests that using the AHSO we were able to train dogs to alert to human specific volatiles and ignore any "non-human" odor. There are hundreds or even thousands of volatiles organic compounds emanating from the human body [20]. These come from different parts of the body (e.g., saliva, skin, breath etc.) and vary between individuals depending on sex, diet, genetics, and health and physiological status [30]. Thus, no two individuals have the same volatile profile and each person has their own and specific odor fingerprint [30]. Despite this, the data showed that dogs were able to alert to five novel participants immediately with no additional training, indicating sufficient common features across volunteers.

To further confirm that the olfactometer delivered an odorant representative of human scent (HS), in Experiment 2, we evaluated certified search and rescue (SAR) dogs with the AHSO. These dogs were naive to the olfactometer but proficient detecting HS in different contexts and were trained independently of the research team. The high performance of SAR dogs when inserting their nose to the odor ports in the AHSO indicated that the apparatus was delivering an accurate representation of human scent and that dogs naïve to the apparatus but trained to detect human specific volatiles were able to detect and alert to human scent in the AHSO. The slight decrement in the proportion of correct responses, sensitivity, and specificity observed in the AHSO relative to the control barrel search was mostly due to the novelty of the task and because the control search did not contain blank trials or distractor odors. Nevertheless, dogs' sensitivity and specificity in the AHSO was > 95%.

The goal of Experiment 3 was to use the AHSO to understand what were the important constituents of human scent dogs were utilizing to make a response. The main advantage of the AHSO is that dogs were trained to whole human scent rather than a single constituent of human scent like breath, or a scented article. Whole human scent is thought to be a combination of breath, skin secretion, and body fluids (i.e., sweat) [12]. Thus, by enclosing a volunteer in an acrylic chamber we trained dogs to detect the headspace of the chamber that should contain volatiles present in the volunteers' breath, skin, and body fluid as a whole. Using a similar acrylic chamber for whole human scent sampling, Rankin-Turner and McMeniman [12] identified over 700 human specific volatiles. Even when they found that there was significant variation between individuals, overall they were able to identify 43 volatiles present in most or all of the 20 volunteers sampled [12]. These 43 volatiles were mostly ketones, terpenes, aldehyde,

and hydrocarbons and all were thought to be of breath and/or skin origin. Based on these findings we can infer that our dogs were trained to detect a complex mixture of human specific volatiles of both breath and skin origin rather than just one component (i.e., breath or skin secretion).

To dissect which was an important constituent of whole human scent that dogs were detecting, we tested dogs' spontaneous response or generalization to different scented articles. After training with whole human scent, dogs showed poor generalization to all the different impregnation methods and substrates used. The response rate to all testing odors with the exception of breath and the positive control were not different from their relevant control suggesting that their exposure to human scent did not increase the probability of alert. This suggest that the materials, the methods, or a combination of both were not able to capture and/or release a combination of volatiles or the concentration of volatiles necessary to present dogs with an odor that was perceptually similar to whole human scent.

For instance, even when Rankin-Turner and McMeniman [12] found hundreds of volatiles in the headspace of the chamber, a previous study only found a total of 58 volatile compounds when a direct extraction of the hand was performed with different textiles [21]. This significant difference in the number of volatiles present in scented textiles and in the headspace of the chamber could result in perceptual differences between samples and thus a low response rate from all dogs. Another possibility could be that the different textiles used for direct extraction did not capture or release the same concentration present in the headspace of the chamber. As noted by other studies from the authors, a possible change in the concentration would have led to reduced generalization [31,32].

Overall dogs showed the highest spontaneous response to the positive control and to breath. Five of the six dogs alerted to breath during the first trial but their response during the remaining trials varied. This suggests that breath is an important constituent of whole human scent but that dogs still showed a significant reduction in the response rate when it was presented individually. Dogs also showed high response rates to the positive control with a novel volunteer. This indicates that dogs were able to respond to a novel scent if it was different, but perceptually similar to whole human scent. This confirms that spontaneous response or generalization to a perceptually similar odor was achievable with the paradigm used. The high response rate to the positive control further confirms that the poor response rate to the other testing odors was due to perceptual differences and not due to the paradigm.

Although not statistically significant, dogs' response rate to the positive control and to breath decreased with time. For instance, dogs' response rate during the second testing session was lower than during the first testing session for both. This likely occurred due to dogs learning the differences between the target and the testing odor. For logistical reasons the target and the testing odor for just these two tests were different volunteers. Existing literature shows variations between individuals scent and breath [12,15]. Furthermore, previous research has also found that dogs can discriminate articles scented by different individuals [33,34] and breath samples of the same individual when stressed [35]. Thus, the reduction in the response rate to breath and the positive control during the second session maybe due to dogs learning the perceptual differences between the target and the testing odor across the repeated generalization tests.

Experiment 3 further suggests that all of our human scent collection methods failed to adequately represent whole human scent, but breath did show an important positive result, indicating that in a search and rescue scenario where a person is confined, breath is one of the more important signatures of human scent.

Experiment 4 replicated and extended the importance of breath as a constituent of HS. In this experiment, by evaluating breath alone, the volunteer with their breath (positive control),

and volunteer with their breath exhausted (Human No Beath), we observed that breath alone was sufficient for spontaneous generalization and that exhausting the breath of the same volunteer significantly reduced generalization. Therefore, Experiment 4 replicated the primary results in Experiment 3. Although the source and the exact combination of volatiles that dogs learn to identify when trained to detect human scent still unknown, a hypothesis accepted by most trainers, scientist, and experts in the field is that human scent is constituted of continuous shedding of skin cells mixed with sweat and other body gland secretions, and bacterial metabolites [3,36]. This is often called the raft detection theory and it proposes that dogs can detect the continuous shedding of skin. This hypothesis suggests that humans are constantly shedding small pieces of skin debris, also known as corneocytes, and that dogs perceive the corneocytes through their olfactory system [3]. For instance, this model will suggest that in a SAR scenario a dog will navigate the environments following disturbances in the terrain and the corneocytes scent until locating the missing person. Recently, Eckenrode et al. [36] proposed a Permeative and Aerosolized Corneocyte (PAC) model for human scent detection. Readers can find a detailed description and justification of this complex model elsewhere [36], but in essence the model proposes that humans shed thousands of corneocytes per second. Corneocytes act as a mesh that captures volatiles of skin origin and serve as bacteria substrate. Dogs' perception of human scent is then a result of the aerosolization of the corneocytes, the volatiles entrained in its keratin like structure, and volatiles produced by the different secretory glands in the skin.

Our results in Experiment 3 & 4 rather highlight that it would be valuable for further scientific research to consider the potential importance of breath associated volatiles. Particularly given that in Experiment 4, the volunteer with no breath led to only a 50% alert rate (compared to 97% for the positive control) whereas breath alone (without the person in the chamber) led to an 88% alert rate. The notion that volatiles of breath origin are an important stimulus in dog perception of human scent could very well be attributed and corroborated with the high number of volatile organic compounds identified in human exhaled breath compared to other human biospecimens when evaluating the human Volatilome [37].

In their literature review about SAR dogs, Jones et al. [3] explained that the odor profile of a human will change almost immediately after death and that even when humans will not perceive immediate changes in odor when a person dies, a dog trained to find live humans will perceive it immediately. They attribute these rapid changes in the scent of a live human from a recently dead person to the loss of aerobic metabolism. This anecdotal observation about detection dogs also suggests that breath is an important constituent of human scent.

Importantly, our results do not indicate that a dog trained to detect a live person is exclusively alerting to the presence of breath. Although not statistically significant, the probability of alert to breath still was almost 10% lower than the probability of alert to whole human scent. Further, dogs did still alert to the person even when breath was exhausted at a rate of 50%, which was higher than any of our controls or other tests in Experiment 3. Instead, our results confirm that from a detection dog perspective, human scent is a combination of volatiles of breath and skin origin. Furthermore, in our laboratory set-up, where a whole human was confined, volatiles from breath were more salient than volatiles from other origins. This suggests that current models of human scent detection should be adapted and reconsider the importance of breath to existing models.

There are some inherent limitations to this research. 1) Experiments were conducted in a laboratory setting and the task does not resemble a SAR search task. 2) Our results are likely limited to a confined human scenario. In contexts outside of this, such as tracking/trailing, it is likely dogs leverage different odors based on the availability of human-associated odors. For example, if dogs are trained to find scented articles, or to track persons, dogs are being trained

to specific constituents of human scent, or items associated with human scent, that would be preserved in ground environments. In such cases, breath could likely be irrelevant because the diffusion of these volatiles might be different and may not be retained in the environment. Our purpose herein was not to model all situations in which a dog could find a person, but rather to investigate the scenario in which a dog is trained to find a trapped and confined person. Within this case, nonetheless, our results highlight that volatiles of breath origin are highly salient odor stimuli for dogs. The AHSO settings is a good laboratory model that resembles a scenario where a person is trapped under a rubble pile in the case of a natural (i.e., tornado, earthquake) or manmade (i.e., explosion) disaster.

Another potential limitation to the AHSO was that the acrylic casing used to trap humidity could have also trapped some water-soluble volatiles. Because condensed water in the acrylic casing was almost never observed, and because the air pump was located after the acrylic casing, this likely had limited impact on the volatiles presented to the dogs. Future studies are needed to confirm our findings with certified SAR dogs in an operational environment. Nevertheless, our results highlight the importance of taking into account volatiles of breath origin when creating possible training aids or generating models to explain detection dog perception of human scent for scenarios in which there is a confined human.

## Supporting information

**S1 File. Experiment 1 data.**
(CSV)

**S2 File. Experiment 2 data.**
(CSV)

**S3 File. Experiment 2 behavior data.**
(CSV)

**S4 File. Experiment 3 data.**
(CSV)

**S5 File. Experiment 4 data.**
(CSV)

**S6 File. Experiment 2 barrel search.**
(CSV)

## Author Contributions

**Conceptualization:** Edgar O. Aviles-Rosa, Paola A. Prada-Tiedemann, Jenna D. Gadberry, Nathaniel J. Hall.

**Data curation:** Edgar O. Aviles-Rosa.

**Formal analysis:** Edgar O. Aviles-Rosa.

**Funding acquisition:** Paola A. Prada-Tiedemann, Michele N. Maughan, Jenna D. Gadberry, Robin R. Greubel, Nathaniel J. Hall.

**Investigation:** Edgar O. Aviles-Rosa.

**Methodology:** Edgar O. Aviles-Rosa, Nathaniel J. Hall.

**Project administration:** Edgar O. Aviles-Rosa.

**Resources:** Robin R. Greubel, Nathaniel J. Hall.

**Supervision:** Edgar O. Aviles-Rosa.

**Validation:** Edgar O. Aviles-Rosa.

**Visualization:** Edgar O. Aviles-Rosa.

**Writing – original draft:** Edgar O. Aviles-Rosa.

**Writing – review & editing:** Edgar O. Aviles-Rosa, Andrea C. Medrano, Ariela Cantu, Paola A. Prada-Tiedemann, Michele N. Maughan, Jenna D. Gadberry, Robin R. Greubel, Nathaniel J. Hall.

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
