## [Decision Letter · Decision Letter 0]

13 Oct 2023

PONE-D-23-25721Development of an Automated Human Scent Olfactometer and its Use to Evaluate Detection Dog Perception of Human ScentPLOS ONE

Dear Dr. Aviles-Rosa,

Thank you for submitting your manuscript to PLOS ONE. After careful consideration, we feel that it has merit but does not fully meet PLOS ONE’s publication criteria as it currently stands. Therefore, we invite you to submit a revised version of the manuscript that addresses the points raised during the review process.

We look forward to receiving your revised manuscript.

Kind regards,

Tommaso Lomonaco, Ph.D

Academic Editor

PLOS ONE

Journal Requirements:

Additional Editor Comments:

Dear Authors, please ready carefully the points raised by the reviewers.

Reviewers' comments:

Reviewer's Responses to Questions

**Comments to the Author**

1. Is the manuscript technically sound, and do the data support the conclusions?

Reviewer #1: Yes

Reviewer #2: No

2. Has the statistical analysis been performed appropriately and rigorously? 

Reviewer #1: Yes

Reviewer #2: Yes

3. Have the authors made all data underlying the findings in their manuscript fully available?

Reviewer #1: Yes

Reviewer #2: Yes

4. Is the manuscript presented in an intelligible fashion and written in standard English?

Reviewer #1: Yes

Reviewer #2: No

5. Review Comments to the Author

Reviewer #1: This is an excellent paper which describes the use of Automated Human Scent Olfactometer for evaluating detection dog perception of human scent. It is highly evaluated that authors addressed the unveiled research question; what the main constituent of the human scent that detection dogs are utilizing to locate a person. I believe the developed system shown in this paper will pave the ways to solve the research question. So, I recommend this paper should be published after minor considerations.

Major points

1) As for the AHSO system, I have a simple question on the influence of the humidity trap which removes excess water on chemical components and/or human scent, because there are water soluble gases in breath and from skin surface. For example, the dermal emission of ammonia is known to cause unpleasant body odor. So, please discuss on this point.

2) Line 1278-1284: Authors described the volatiles of breath origin are the main constituent of human scent. I can agree to this statement based on the results in these experiments. However, the breath gases and skin volatiles were well mixed and served for dogs after removing excess water in this system. In actual environment, dogs near the ground may not be exposed to a combination of both volatiles of breath and skin origin, because the diffusion of these gases can be different. So, authors should add the limitation on this point.

Minor points

1) Abstract: please describe the “HS” in full spelling.

2) L60-61: As described above, volatile inorganic compounds such as ammonia and hydrogen sulfide are also contribute to human scent.

Reviewer #2: 1. The manuscript “Development of an Automated Human Scent Olfactometer and its Use to Evaluate Detection Dog Perception of Human Scent”, authored by Dr. Aviles-Rosa et al., describe the authors efforts to try to determine “what constitutes human scent from a detection dog perspective”. In short, it is an experimental attempt to train dogs to forget about their natural tendency to use discrimination in favor of generalization when tracking something, humans in this case (Moser AY et al., doi: 10.3390/ani9090702). The research problem, however, conveys an extremely difficult question to answer for us humans, because dogs have the uncanny ability to discriminate one human from another even if they are genetically identical, eat the same food, and have lived all their lives under the same environment, as has been solidly demonstrated with monozygotic twins (Pinc L et al., doi: 10.1371/journal.pone.0020704). It means that, for the authors to be able to find the scent information used by dogs to identify any and all human beings as a single species (as they do to find surviving earthquake victims, for instance), the scientists would have to use many human subjects to try to represent, at the very least, a minimal approximation to the smell that dogs track when looking for any and all humans instead of a single individual within Homo sapiens. This shortcut is serious because the study is severely underpowered. Dogs do generalize very easily a particular scent like that of a person infected by SARS-CoV-2, and are able to identify any human infected by it from within many healthy individuals in a fraction of a second and under a wide variety of circumstances. That is easy for them, because in nature canids survive by discriminating the scent of their specific pray even if it is part of large herd, and it makes them extraordinarily efficient at discriminating. In this study, the authors tried to train dogs to do the opposite, i.e., to identify any individual from the human herd of 8 billion individuals. Although dogs and many other animals identify any human scent as generically “human” since very early in life, they always discriminate between them, and using only a handful of human volunteers to train them might fail to let the dogs know what the researcher want from them. Beyond power, the authors did not use the appropriate methodology to train the dogs to tell the trainer when they were generalizing for the human scent and when they were discriminating one human from another (see below). And finally, the 4 experiments were not sufficient to answer the research question, and generated conflicting results that the authors were unable to explain in their quite extensive Discussion. These methodological flaws prevent the publication of this piece of work in PLOS ONE.

2. Experiment 1 was designed to determine if 2 dogs “can successfully be trained to identify the headspace from the chamber that held the human volunteer (n = 10) and discriminate it from the headspace of both distractor chambers and various distractor odors including personal hygiene products”. The researchers achieved the first goal (i.e., both dogs were trained to identify which one of three chambers was holding the human volunteer) but failed to train both dogs to discriminate between three empty chambers after erasing the odor of the human volunteer (see “Control group” in Fig. 1). Contrary to the author’s interpretation (line 845-847: “This indicates […] that dogs were responding to the presence of these human specific volatiles”), the result of Experiment 1 tells me that the dogs recurred to false-positive indications when all three chambers where empty, a virtually constant problem when dogs are trained to detect the odor instead of its source.

3. Experiment 2 involved 5 experienced search and rescue (SAR-certified) dogs “to test if [they] were able to identify the headspace of the chamber containing the human volunteer in the olfactometer”. Only 5 of 8 SAR-certified dogs were trained successfully to do the scent work with the apparatus. All 5 performed almost flawlessly and detected the presence of human odor (apparatus) or of an actual human being (barrels) correctly 85% and 92% of the time, respectively. The 7% difference was not significant, demonstrating that SAR dogs are equally adept at finding the source of human scent directly (barrel) or indirectly (apparatus), and that the apparatus designed by the researchers was efficient for the specific task of carrying the scent of the human subject occupying the chamber. If the procedure to eliminate human odor from the apparatus was efficient, the results of Experiment 2 also suggest that failure in Control Tests during Experiment 1 depended on the dogs’ training.

4. Experiment 3 was made with 6 new dogs that had failed certification as explosive detectors (the authors re-trained these dogs with the apparatus). The goal was to “understand what was the main constituent of human scent dogs were utilizing to make a response”, which was also the main objective of the study. Experiment 3 involves too many scenting scenarios, but all of them fulfill one of three scent sources: the whole target or its odor (one of 4 human volunteers), a part of the target or its odor (breath, or an article scented with a human’s axilla, foot, torso, hair, or their combination), or no target at all (blanks). As an additional confusing variable (for the canines), the authors did not inform the dogs when their alerts were correct in a significant fraction of the trial. Figure 7 shows that the probability of alert went from higher to lower (>) in this direction: first, the target (one human in the chamber) > second, the positive control (a different human in the chamber) > third, the breath of the targeted human > any other odor taken from the human source. The difference between the target and the positive control shows that the dogs were discriminating one human from another instead of generalizing for all humans, and the difference between human breath and the other sources of human odor suggest that no odor was present for the dogs. Among several explanations for such results, two seem to be the most probable: one, contrary to human’s breath, the apparatus did not carry to the dogs the odor taken from humans’ skin, and two, the training was inadequate to teach the dogs to alert when scenting the odor of any human, independently of the volatile organic compounds presented to them. These results do not support the author’s conclusion that breath is the “key component” used by dogs to scent-find a human being, and it led to Experiment 4.

5. Experiment 4 included the same 6 dogs from Experiment 3 but only 2 human volunteers, and the authors did not reward the dogs at all (one dog refuse to work under this unrewarding conditions). The main objective was to compare the correct alert rate to a human inside the chamber with the same subject inside the chamber respiring into a scuba equipment, thus eliminating breath from the scent information. The 5 remaining dogs alerted correctly only 50% of the time when breath was not part of the scent information, and the authors thought that it confirmed the hypothesis that breath was the key component of human scent used by dogs to identify us. My interpretation of the data is that breath provided the dogs with part of the scent information they were sniffing for, and eliminating it decreased their performance proportionately. Also, it has been demonstrated that eliminating reward affects negatively the dogs’ performance, as was evident in Experiments 3 and 4 as the scent tasks became more demanding: Cimarelli G et al., doi: 10.1007/s10071-020-01425-9; and Peiris PL et al., doi: 10.1002/jeab.790.

Next, I will enumerate minor observations to the manuscript, followed by my comment in red font:

6. Line 61: Different attempts have been made to characterize human scent utilizing different analytical techniques and sensors (30–39) but none of these attempts included the use of detection dogs. This sentence does not consider relevant research in the area (reviewed by Angle et al., doi: 10. 3389/fvets.2016.00047).

7. Line 99: The PTFE tubing from each olfactometer was connected to an odor port. IR beam sensors were located at the front of each port. Please define any abbreviation the first time you use it in the manuscript (Poly Tetra Fluoro Ethylene, or Teflon® tubing). The same rule applies to all other abbreviations, like HS, IR, PVC, RH, RO, etc. For instance, human scent (HS) is used first in the abstract, and many-times thereafter before being described in line 169.

8. Line 98: Figure 1. A. The schematic of the Automated Human Scent Olfactometer (AHSO). B. A picture of the dog interface. Line 98 contains the texto of Figure 1, but Figure 1 has not been cited yet. Therefore, it appears again in line 149.

9. Line 103: acrylic chamber.shows a schematic of the AHSO. This sentence in line 103 makes no sense and is not separated from the text of Figure 1.

10. Line 174: The beginning of a trial was marked with a trial initiation tone. Lines 177 & 178: If dogs alerted to the correct port, the computer program marked the response with a “bleep” sound and the experimenter reinforced the alert with a treat. I suppose these tones and sounds were audible to the dog, but it is better if the authors make it clear.

11. The authors opted to log-transform the data to apply parametric statistics instead of analyzing actual data with non-parametric statistics. Log-transforming the data helps them to find statistical differences, but the number of dogs is so small that it makes such parametric analysis hardly reliable. They should use non-parametric tools for the statistical analysis of untransformed data.

12. Most experts believe that hundreds of human volunteers are required to validate this kind of data, with the argument that dogs quickly learn by memory the odor of each of the human participants. The dogs knew when the authors changed the human subject, and it took them just a couple of exposures the discriminate between the very small number of human volunteers. There is no way for the authors to know if incorrect alerts did obey to the dogs discriminating between the different volunteers.

13. The paper is written in a very confusing order. The first 3 figures and table appear suddenly without any citation to them whatsoever.

14. The paper is extremely long. The Methods could be shortened significantly if, instead of describing each experiment by separate, the author put together the aspects common to all four experiments and then, in a single paragraph, point to their individual characteristics. The Discussion must be shortened to just 4-5 paragraphs. It is not necessary to try to explain every conflicting result with speculation, but to focus on the actual methodological problems and their solutions for future research.

6. PLOS authors have the option to publish the peer review history of their article (what does this mean?). If published, this will include your full peer review and any attached files.

Reviewer #1: **Yes: **Yoshika Sekine

Reviewer #2: No

---

## [Author Response · Author response to Decision Letter 0]

27 Nov 2023

Reviewer #1: 

This is an excellent paper which describes the use of Automated Human Scent Olfactometer for evaluating detection dog perception of human scent. It is highly evaluated that authors addressed the unveiled research question; what the main constituent of the human scent that detection dogs are utilizing to locate a person. I believe the developed system shown in this paper will pave the ways to solve the research question. So, I recommend this paper should be published after minor considerations.

Thank you for taking the time to review our manuscript. 

Major points

1) As for the AHSO system, I have a simple question on the influence of the humidity trap which removes excess water on chemical components and/or human scent, because there are water soluble gases in breath and from skin surface. For example, the dermal emission of ammonia is known to cause unpleasant body odor. So, please discuss on this point.

This is a great observation by the reviewer. Yes, water condensation in the acrylic casing could potentially have trapped water soluble volatiles. Although we use the acrylic casing to reduce humidity from entering and contaminating the system, we never observed a significant amount of water condensation in it. Furthermore, because of the way the system was built, dogs were also exposed to volatiles that might have been in the humidity trap since the vacuum pump was after the trap and carried air from the humidity trap (See Figure 1). Thus, although we agree that the humidity trap could have trapped some water-soluble volatiles, based on the small amount of water captured by it, the position of the pump within the system, we think that it had low impact trapping water soluble volatiles. We added the following text to the discussion to address this. 

 “Another potential limitation to the AHSO was that the acrylic casing intended to trap humidity could have also trapped some water-soluble volatiles. Because condensed water in the acrylic casing was almost never observed, and because the air pump was located after the acrylic casing pulling odor from the casing, this likely had limited impact on the volatiles presented to the dogs.”

2) Line 1278-1284: Authors described the volatiles of breath origin are the main constituent of human scent. I can agree to this statement based on the results in these experiments. However, the breath gases and skin volatiles were well mixed and served for dogs after removing excess water in this system. In an actual environment, dogs near the ground may not be exposed to a combination of both volatiles of breath and skin origin, because the diffusion of these gases can be different. So, authors should add the limitation on this point.

Another great point by the reviewer. We now address this point in the manuscript where we state that our results are limited to dogs trained to find humans in confined scenarios (i.e., inside cars, under rubble piles, etc.). The following was added to the discussion. 

“Our results are likely limited to a confined human scenario. In contexts outside of this, such as tracking/trailing, it is likely dogs leverage different odors based on the availability of human-associated odors. For example, if dogs are trained to find scented articles, or to track persons, dogs are being trained to specific constituents of human scent, or items associated with human scent, that would be preserved in ground environments. In such cases, breath could likely be irrelevant because the diffusion of these volatiles might be different and may not be retained in the environment.”

Minor points

1) Abstract: please describe the “HS” in full spelling.

This was added to the manuscript.

2) L60-61: As described above, volatile inorganic compounds such as ammonia and hydrogen sulfide are also contribute to human scent.

Thank you for this observation. This line was rephrased as follows:

Human scent (HS) is a complex blend of volatile organic and inorganic compounds (VOCs) of skin, breath, and bodily fluids origin. 

Reviewer #2: 

1. The manuscript “Development of an Automated Human Scent Olfactometer and its Use to Evaluate Detection Dog Perception of Human Scent”, authored by Dr. Aviles-Rosa et al., describe the authors efforts to try to determine “what constitutes human scent from a detection dog perspective”. In short, it is an experimental attempt to train dogs to forget about their natural tendency to use discrimination in favor of generalization when tracking something, humans in this case (Moser AY et al., doi: 10.3390/ani9090702). The research problem, however, conveys an extremely difficult question to answer for us humans, because dogs have the uncanny ability to discriminate one human from another even if they are genetically identical, eat the same food, and have lived all their lives under the same environment, as has been solidly demonstrated with monozygotic twins (Pinc L et al., doi: 10.1371/journal.pone.0020704). It means that, for the authors to be able to find the scent information used by dogs to identify any and all human beings as a single species (as they do to find surviving earthquake victims, for instance), the scientists would have to use many human subjects to try to represent, at the very least, a minimal approximation to the smell that dogs track when looking for any and all humans instead of a single individual within Homo sapiens. This shortcut is serious because the study is severely underpowered. Dogs do generalize very easily a particular scent like that of a person infected by SARS-CoV-2, and are able to identify any human infected by it from within many healthy individuals in a fraction of a second and under a wide variety of circumstances. That is easy for them, because in nature canids survive by discriminating the scent of their specific pray even if it is part of large herd, and it makes them extraordinarily efficient at discriminating. In this study, the authors tried to train dogs to do the opposite, i.e., to identify any individual from the human herd of 8 billion individuals. Although dogs and many other animals identify any human scent as generically “human” since very early in life, they always discriminate between them, and using only a handful of human volunteers to train them might fail to let the dogs know what the researcher want from them. Beyond power, the authors did not use the appropriate methodology to train the dogs to tell the trainer when they were generalizing for the human scent and when they were discriminating one human from another (see below). And finally, the 4 experiments were not sufficient to answer the research question, and generated conflicting results that the authors were unable to explain in their quite extensive Discussion. These methodological flaws prevent the publication of this piece of work in PLOS ONE.

We appreciate the Reviewer’s time and comments. We agree with many of the important capabilities pointed out by the reviewer. However, it is also true that dogs have the ability to generalize to similar stimuli and this has been reported in many studies (Aviles-Rosa et al. 2020; Range et al., 2008; Hall et al 2018; DeGreef et al. 2020) including the one cited by the reviewer (Moser et al. 2019). The existing literature suggests that generalization and discrimination is a complex topic, and it is overly simplistic to say that dog’s natural tendency is to discriminate rather than generalize. Existing literature in humans and animals show that the shape of the generalization curve of an individual (e.g., individuals’ tendency to generalize or discriminate) is influenced by many factors and can be modified through experience and training. 

We feel that the primary criticisms raised are an unfair assessment of this work because: 1) our study leveraged over 12 different human-participants as scent sources, which exceeds that of many published literature we are aware of on this topic, 2) dogs did not show any decrement in alerting to a novel person when a novel person was used as a target (e.g., Figure 4) indicating that dogs spontaneously generalized across sessions when the human scent source was changed, 3) we only saw discrimination between human participants in Experiment 3 when the different participants were being compared within a session and only after multiple non-reinforced generalization trials, which was then remedied with a within-participant design in Experiment 4 where no discrimination was observed between positive controls, with the primary results being replicated across the two experiments. Given the robust statistical effects observed, large effect sizes, spontaneous generalization across the 12 human scent sources in Experiment 1, the within-study replication of results using two different experimental paradigms, we disagree that the study is underpowered to yield publication worthy results. 

Based on additional comments, we have made many substantial changes to the manuscript, which are detailed below. 

2. Experiment 1 was designed to determine if 2 dogs “can successfully be trained to identify the headspace from the chamber that held the human volunteer (n = 10) and discriminate it from the headspace of both distractor chambers and various distractor odors including personal hygiene products”. The researchers achieved the first goal (i.e., both dogs were trained to identify which one of three chambers was holding the human volunteer) but failed to train both dogs to discriminate between three empty chambers after erasing the odor of the human volunteer (see “Control group” in Fig. 1). Contrary to the author’s interpretation (line 845-847: “This indicates […] that dogs were responding to the presence of these human specific volatiles”), the result of Experiment 1 tells me that the dogs recurred to false-positive indications when all three chambers where empty, a virtually constant problem when dogs are trained to detect the odor instead of its source.

We apologize for the confusion here and have edited the manuscript in this section to improve clarity. 

In Experiment 1, in the final training sessions, false alerts occurred on 1.5% of trials, indicating that dogs rarely alerted on the distractor chambers, which is well within the limits of prior research and is within certification standards for detection dogs (e.g. explosive standards: https://www.aafs.org/sites/default/files/media/documents/092_Std_e1.pdf). 

Additionally, we have revised the manuscript to clarify the results of the negative control test. The purpose of this test was to confirm that dogs could not solve the discrimination due to any unintentional system differences/olfactometer cues, such as solenoid valve sounds differences in air flow, differences in air pressures, etc. This type of test is standard in olfactometry research (e.g., Bodyak & Slotnick, 1999) and has been extensively used in our lab to confirm dogs are not leveraging unintentional cues (e.g., Aviles-Rosa, DeChant, et al., 2023; Aviles-Rosa et al., 2021, 2022; Aviles-Rosa, Nita, et al., 2023; M. DeChant et al., 2023; M. T. DeChant et al., 2023; Hall et al., 2013, 2014, 2015, 2016; Hall & Wynne, 2018). 

Furthermore, we have revised the results of these tests for clarity. Briefly, the control test consisted of 10 trials where no person was placed in any chamber. To keep consistent with experimental trials, 8 of the ten trails activated the air from the chamber that formerly held the human-participant. Two trials remained the “blanks” and did not present the odor from that chamber. A correct response was scored identically to an experimental session (to make direct comparisons), even though we did not purposefully present human scent. Our results showed that if everything is conducted identically to an experimental session with the exception of the presence of a human participant, dogs were not able to correctly identify the headspace of the chamber that formerly held the participant after cleaning. The 20 and 25% of correct responses indicate that the only correct responses that occurred were during the 2 blank trials. Out of the 40 control trials (20 trials in phase 2 and 20 trials at the end of the experiment) dogs false alerted only in 2 trials (e.g., 5%) and made an “all clear” response on 38 of these trials. This is well within the false alert rate acceptable for detection dog standards (https://www.aafs.org/sites/default/files/media/documents/092_Std_e1.pdf). 

3. Experiment 2 involved 5 experienced search and rescue (SAR-certified) dogs “to test if [they] were able to identify the headspace of the chamber containing the human volunteer in the olfactometer”. Only 5 of 8 SAR-certified dogs were trained successfully to do the scent work with the apparatus. All 5 performed almost flawlessly and detected the presence of human odor (apparatus) or of an actual human being (barrels) correctly 85% and 92% of the time, respectively. The 7% difference was not significant, demonstrating that SAR dogs are equally adept at finding the source of human scent directly (barrel) or indirectly (apparatus), and that the apparatus designed by the researchers was efficient for the specific task of carrying the scent of the human subject occupying the chamber. If the procedure to eliminate human odor from the apparatus was efficient, the results of Experiment 2 also suggest that failure in Control Tests during Experiment 1 depended on the dogs’ training.

In Experiment 2 we conducted a positive control which is different from the negative control test done in Experiment 1. We have edited the manuscript to clarify this and to have more consistent labeling for the various control tests implemented through the manuscript. 

 Negative and positive controls are two different procedures and the expected outcome of both are different. While in a negative control the desired outcome is a significant decrement in performance, the opposite is expected in a positive control. In the positive control case, statistically similar results in the barrel search and the AHSO indicate that SAR dogs were able to identify HS in the AHSO as effectively as in the barrel search. This further confirms that the AHSO was delivering an odor that was perceptually similar to the odor of a human in a barrel search. 

4. Experiment 3 was made with 6 new dogs that had failed certification as explosive detectors (the authors re-trained these dogs with the apparatus). The goal was to “understand what was the main constituent of human scent dogs were utilizing to make a response”, which was also the main objective of the study. Experiment 3 involves too many scenting scenarios, but all of them fulfill one of three scent sources: the whole target or its odor (one of 4 human volunteers), a part of the target or its odor (breath, or an article scented with a human’s axilla, foot, torso, hair, or their combination), or no target at all (blanks). As an additional confusing variable (for the canines), the authors did not inform the dogs when their alerts were correct in a significant fraction of the trial. Figure 7 shows that the probability of alert went from higher to lower (>) in this direction: first, the target (one human in the chamber) > second, the positive control (a different human in the chamber) > third, the breath of the targeted human > any other odor taken from the human source. The difference between the target and the positive control shows that the dogs were discriminating one human from another instead of generalizing for all humans, and the difference between human breath and the other sources of human odor suggest that no odor was present for the dogs. Among several explanations for such results, two seem to be the most probable: one, contrary to human’s breath, the apparatus did not carry to the dogs the odor taken from humans’ skin, and two, the training was inadequate to teach the dogs to alert when scenting the odor of any human, independently of the volatile organic compounds presented to them. These results do not support the author’s conclusion that breath is the “key component” used by dogs to scent-find a human being, and it led to Experiment 4.

We identify three primary concerns from the reviewer: 1) generalization testing occurred under an intermittent schedule of reinforcement 2) dogs showed a little discrimination between the reinforced target and the positive control probe in Experiment 3, and 3) the olfactometer did not present human skin odor. We address these concerns in order. 

1) Dogs were accustomed to an intermittent schedule of reinforcement prior to testing to allow for repeated generalization testing. This is critical to allow for us to evaluate dogs’ spontaneous response to a presented odor without explicitly reinforcing or “training” responses to all odors. We never claim that dogs cannot be trained to respond to samples previously in contact with human skin (this can certainly be done), but rather the question is whether dogs spontaneously respond to these kinds of stimuli when trained to the scent from an entire, enclosed, human. This allows us to then understand the relative salience of different components of “human scent”.

Intermittent schedules of reinforcement followed by testing under extinction are the standard method for assessing spontaneous generalization in animals (for a review see Dinsmoor, 1995; for a classic example, see Guttman, 1959) and are suggested for use in detection canine evaluation by other lab groups (Lazarowski et al., 2020) and have been previously established as a generalization protocol in our laboratory (Aviles-Rosa et al., 2022). We have clarified our goals throughout the manuscript to assess spontaneous generalization and now better justify our use of the intermittent schedule of reinforcement. 

2) In Experiment 3, different human scent volunteers were used between the target person (the reinforced response) and the positive control and breath (generalization samples tested under extinction). This was due to logistical reasons (a person could not be in a box and providing breath at the same time or in two boxes at the same time). All other test samples came from the same individual as the target person. Thus, any decrement in the remaining test samples is not because the dogs are “discriminating” vs “generalizing” between people, the dogs are failing to respond to the sample from the same person. 

a. Importantly, a lack of response does not indicate that no odor was available. Our paradigm was designed to assess spontaneous generalization, and we were therefore measured how similar one odor was to the whole human scent, not whether a dog could be trained to detect that odor. This is clarified in the discussion. 

b. Five out of the six dogs responded to the positive control on the first trial (added to the manuscript), indicating that most dogs did spontaneously generalize to the other person without issue, and only showed discrimination after repeated non-reinforced generalization trials. This was the expected result which is necessary for spontaneous generalization assessments (Aviles-Rosa et al., 2022; Dinsmoor, 1995; Lazarowski et al., 2020).

c. Even though dogs showed a small difference between the target person and the positive control, dogs still showed a strong response to the breath of a different person. Thus, making the impact of breath stronger not weaker. Thus, in spite of some discrimination between the two humans, dogs still responded most to the beath. Thus, this difference strengthens the argument that breath is important. 

d. Experiment 4 assessed generalization to all test stimuli using the same human as the target and donor of odor constituents. In this case, dogs showed no discrimination between the target and positive control. Further, these comparisons were entirely using the same human scent volunteer. Thus, these results could not be due to dogs discriminating between different human scent volunteers. Thus, even when evaluating generalization from a specific human to the constituents of that same person, the results from Experiment 3 were replicated. 

3) We agree that it is possible that properties of odor emitted from the skin may not have traveled as well for odor presentation in the olfactometer- perhaps due to solubility of these molecules. We have now included such a limitation in the discussion (copied below). Nonetheless, this is not a critical limitation for our results, because if skin volatiles cannot move/transport as effectively as breath volatiles through active air transport, it remains unlikely that dogs trained to find humans in search and rescue scenarios (such as trapped survivors) would be able to leverage the molecules that show such poor transport/movement dynamics. Furthermore, Seach and Rescue canines did readily show odor recognition from the olfactometer, indicating that odor representative of their trained human scent (which was done independent of this project and independent of the olfactometer), was adequately presented by the olfactometer. 

“Another potential limitation to the AHSO was that the acrylic casing used to trap humidity could have also trapped some water-soluble volatiles. Because condensed water in the acrylic casing was almost never observed, and because the air pump was located after the acrylic casing pulling odor from the casing, this likely had limited impact on the volatiles presented to the dogs. Future studies are needed to confirm our findings with certified SAR dogs in an operational environment.”

5. Experiment 4 included the same 6 dogs from Experiment 3 but only 2 human volunteers, and the authors did not reward the dogs at all (one dog refuse to work under this unrewarding conditions). The main objective was to compare the correct alert rate to a human inside the chamber with the same subject inside the chamber respiring into a scuba equipment, thus eliminating breath from the scent information. The 5 remaining dogs alerted correctly only 50% of the time when breath was not part of the scent information, and the authors thought that it confirmed the hypothesis that breath was the key component of human scent used by dogs to identify us. My interpretation of the data is that breath provided the dogs with part of the scent information they were sniffing for, and eliminating it decreased their performance proportionately. Also, it has been demonstrated that eliminating reward affects negatively the dogs’ performance, as was evident in Experiments 3 and 4 as the scent tasks became more demanding: Cimarelli G et al., doi: 10.1007/s10071-020-01425-9; and Peiris PL et al., doi: 10.1002/jeab.790.

We believe our interpretation agrees with the reviewer’s and perhaps the disagreement is semantic. The reviewer’s “[my] interpretation of the data is that breath provided the dogs with part of the scent information they were sniffing for, and eliminating it decreased their performance proportionately”. 

We completely agree, but simply add that if the proportional drop in responding is ~50% when breath is removed, and that the presentation of breath alone leads to an 88% response rate, then that item must be a “key” (i.e., important) constituent. We have changed wording to simply “important” to perhaps reduce disagreement. Nonetheless, we completely agree with the reviewer that results from this experiment do not show that breath was the only source of human scent but rather an important (or “key”) component. Our conclusion from all the experiments was never that breath was the sole source of human scent but rather an important component. We have clarified this in the manuscript and copy below. 

With respect to the reinforcement rate, please see our response #1 in the above comment. 

“Importantly, our results do not indicate that a dog trained to detect a live person is exclusively alerting to the presence of breath. Although not statistically significant, the probability of alert to breath still was almost 10% lower than the probability of alert to whole human scent. Further, dogs did still have alerts to the person even when breath was exhausted at a rate of 50%, which was higher than any of our controls or other tests in Experiment 3. Instead, our results confirm that from a detection dog perspective, human scent is a combination of volatiles of breath and skin origin. Furthermore, in our laboratory set-up, where a whole human was confined, volatiles from breath were more salient than volatiles from other origins. This suggests that current models of human scent detection should at least consider the importance of breath to existing models. 

There are some inherent limitations to this research. 1) Experiments were conducted in a laboratory setting and the task does not resemble a SAR search task. 2) Our results are likely limited to a confined human scenario. In contexts outside of this, such as tracking/trailing, it is likely dogs leverage different odors based on the availability of human-associated odors. For example, if dogs are trained to find scented articles, or to track persons, dogs are being trained to specific constituents of human scent, or items associated with human scent, that would be preserved in ground environments. In such cases, breath could likely be irrelevant because the diffusion of these volatiles might be different and may not be retained in the environment. Our purpose herein was not to model all situations in which a dog could find a person, but rather to investigate the scenario in which a dog is trained to find a trapped and confined person.”

Next, I will enumerate minor observations to the manuscript, followed by my comment in red font:

6. Line 61: Different attempts have been made to characterize human scent utilizing different analytical techniques and sensors (30–39) but none of these attempts included the use of detection dogs. This sentence does not consider relevant research in the area (reviewed by Angle et al., doi: 10. 3389/fvets.2016.00047).

We agree with the reviewer that this is a great paper. We cited the paper in the introduction as suggested. 

7. Line 99: The PTFE tubing from each olfactometer was connected to an odor port. IR beam sensors were located at the front of each port. Please define any abbreviation the first time you use it in the manuscript (Poly Tetra Fluoro Ethylene, or Teflon® tubing). The same rule applies to all other abbreviations, like HS, IR, PVC, RH, RO, etc. For instance, human scent (HS) is used first in the abstract, and many-times thereafter before being described in line 169.

Thank you. We revised the manuscript and defined the abbreviations the first time they appear. 

8. Line 98: Figure 1. A. The schematic of the Automated Human Scent Olfactometer (AHSO). B. A picture of the dog interface. Line 98 contains the texto of Figure 1, but Figure 1 has not been cited yet. Therefore, it appears again in line 149.

Figure 1 is cited in the manuscript in lines 98 and 132 before the figure captures appears in line 144.

9. Line 103: acrylic chamber.shows a schematic of the AHSO. This sentence in line 103 makes no sense and is not separated from the text of Figure 1.

We apologize for this. This must have been an error when creating the PDF package for submission. We will carefully check for this error in the resubmission. 

10. Line 174: The beginning of a trial was marked with a trial initiation tone. Lines 177 & 178: If dogs alerted to the correct port, the computer program marked the response with a “bleep” sound and the experimenter reinforced the alert with a treat. I suppose these tones and sounds were audible to the dog, but it is better if the authors make it clear.

Thank you for this observation. The following was added to the manuscript for clarification. 

Line 169 All tones used by the computer program were audible to the dog.

11. The authors opted to log-transform the data to apply parametric statistics instead of analyzing actual data with non-parametric statistics. Log-transforming the data helps them to find statistical differences, but the number of dogs is so small that it makes such parametric analysis hardly reliable. They should use non-parametric tools for the statistical analysis of untransformed data.

We are unaware of prior research that has demonstrated that log transformation to meet parametric assumptions followed by parametric analyses yields unreliable results. Nonetheless, where we used log transformation of sniff time data in Experiment 2, re-analysis with both parametric and non-parametric results yields the same conclusion- there was no difference in sniff time between the different odors. We prefer to maintain the parametric analysis with the log transformation to keep the parametric analysis procedures consistently applied throughout the manuscript. 

12. Most experts believe that hundreds of human volunteers are required to validate this kind of data, with the argument that dogs quickly learn by memory the odor of each of the human participants. The dogs knew when the authors changed the human subject, and it took them just a couple of exposures the discriminate between the very small number of human volunteers. There is no way for the authors to know if incorrect alerts did obey to the dogs discriminating between the different volunteers.

Thank you for this note. Across our four experiments, 12 volunteers were used. Our review of the literature has found that most studies used a handful of human odor source volunteers: 

Pinc et al. (2011)- 2 monozygotic twins and 2 dizygotic twins (e.g., 8 volunteers)

Prada et al. (2011)- 6 human scent volunteers

Von Durckheim et al. (2018)- 26 human volunteers

Wells & Hepper (2003)- 1 handler volunteer

Brisbin and Austad (1991)- 2 volunteers (dog’s own handler and stranger)

Settle et al., (1994)- 6 volunteer donors

Curran et al., (2010)- 10 volunteers in the scenario

Larger samples sizes are only used when human scent matching studies are conducted (e.g., Jezierski et al., 2010; Marchal et al., 2016), which was not the topic of our experiment.

Regarding discrimination between persons. We would like to highlight that our experiments were purposely designed to avoid discrimination between persons because we were aware of this. To avoid discrimination between persons, we always conducted a within person comparison. For instance, testing odors were impregnated by the same volunteer in the chamber during a testing session. In this way we ensure that lack of responses to the testing odor was not due to discrimination between person but rather due to perceptual differences between the headspace of the chamber with the volunteer and the testing odor impregnated by the same individual.

Please see our response to Comment # 4, 2a-d in regard to concerns raised regarding discrimination between volunteers. 

13. The paper is written in a very confusing order. The first 3 figures and table appear suddenly without any citation to them whatsoever.

We revised the paper and ensured that all figures were cited in the manuscript.

14. The paper is extremely long. The Methods could be shortened significantly if, instead of describing each experiment by separate, the author put together the aspects common to all four experiments and then, in a single paragraph, point to their individual characteristics. The Discussion must be shortened to just 4-5 paragraphs. It is not necessary to try to explain every conflicting result with speculation, but to focus on the actual methodological problems and their solutions for future research.

The methods and discussion were revised extensively cutting over 10 pages from the manuscript, and we deleted sections as suggested.

References

Aviles-Rosa, E. O., DeChant, M. T., Prada-Tiedemann, P. A., & Hall, N. J. (2023). A laboratory model of canine search vigilance decrement, I. Journal of the Experimental Analysis of Behavior.

Aviles-Rosa, E. O., Fernandez, L. S., Collins-Pisano, C., Prada-Tiedemann, P. A., & Hall, N. J. (2022). The use of an intermittent schedule of reinforcement to evaluate detection dogs’ generalization from smokeless-powder. Animal Cognition, 25(6), 1609–1620.

Aviles-Rosa, E. O., Gallegos, S. F., Prada-Tiedemann, P. A., & Hall, N. J. (2021). An Automated Canine Line-Up for Detection Dog Research. Frontiers in Veterinary Science, 8, 775381. https://doi.org/10.3389/fvets.2021.775381

Aviles-Rosa, E. O., Nita, M., Feuerbacher, E., & Hall, N. J. (2023). An evaluation of Spotted Lanternfly (Lycorma delicatula) detection dog training and performance. Applied Animal Behaviour Science, 258, 105816.

Bodyak, N., & Slotnick, B. (1999). Performance of Mice in an Automated Olfactometer: Odor Detection, Discrimination and Odor Memory. Chemical Senses, 24(6), 637–645. https://doi.org/10.1093/chemse/24.6.637

Brisbin, I. L., & Austad, S. N. (1991). Testing the individual odour theory of canine olfaction. Animal Behaviour, 42(1), 63–69. https://doi.org/10.1016/S0003-3472(05)80606-2

Curran, A. M., Prada, P. A., & Furton, K. G. (2010). Canine human scent identifications with post-blast debris collected from improvised explosive devices. Forensic Science International, 199(1–3), 103–108. https://doi.org/10.1016/j.forsciint.2010.03.021

DeChant, M., Aviles Rosa, E. O., Prada‐Tiedemann, P. A., & Hall, N. J. (2023). A laboratory model of canine search vigilance decrement, II: Noncontingent reward and Pavlovian appetitive stimuli. Journal of the Experimental Analysis of Behavior.

DeChant, M. T., Aviles‐Rosa, E., Prada‐Tiedemann, P., & Hall, N. J. (2023). Part, III: Increasing odor detection performance after training with progressively leaner schedules of odor prevalence. Journal of the Experimental Analysis of Behavior.

Dinsmoor, J. A. (1995). Stimulus Control: Part I. The Behavior Analyst, 18(1), 51–68. https://doi.org/10.1007/BF03392691

Guttman, N. (1959). Generalization gradients around stimuli associated with different reinforcement schedules. Journal of Experimental Psychology, 58(5), 335–340. https://doi.org/10.1037/h0045679

Hall, N. J., Collada, A., Smith, D. W., & Wynne, C. D. L. (2016). Performance of domestic dogs on an olfactory discrimination of a homologous series of alcohols. Applied Animal Behaviour Science, 178, 1–6. https://doi.org/10.1016/j.applanim.2016.03.016

Hall, N. J., Glenn, K., Smith, D. W., & L, D. (2015). Performance of Pugs, German Shepherds, and Greyhounds (Canis lupus familiaris) on an odor-discrimination task. Journal of Comparative Psychology, 129(3), 237–246. https://doi.org/10.1037/a0039271

Hall, N. J., Smith, D. W., & Wynne, C. D. L. (2013). Training domestic dogs (Canis lupus familiaris) on a novel discrete trials odor-detection task. Learning and Motivation, 44(4), 218–228. https://doi.org/10.1016/j.lmot.2013.02.004

Hall, N. J., Smith, D. W., & Wynne, C. D. L. (2014). Effect of odor preexposure on acquisition of an odor discrimination in dogs. Learning & Behavior, 1–9. https://doi.org/10.3758/s13420-013-0133-7

Hall, N. J., & Wynne, C. D. L. (2018). Odor mixture training enhances dogs’ olfactory detection of Home-Made Explosive precursors. Heliyon, 4(12), e00947. https://doi.org/10.1016/j.heliyon.2018.e00947

Jezierski, T., Górecka-Bruzda, A., Walczak, M., Świergiel, A. H., Chruszczewski, M. H., Pearson, B. L., & others. (2010). Operant conditioning of dogs (Canis familiaris) for identification of humans using scent lineup. Animal Science Papers and Reports, 28(1), 81–93.

Lazarowski, L., Krichbaum, S., DeGreeff, L. E., Simon, A., Singletary, M., Angle, C., & Waggoner, L. P. (2020). Methodological Considerations in Canine Olfactory Detection Research. Frontiers in Veterinary Science, 7. https://doi.org/10.3389/fvets.2020.00408

Marchal, S., Bregeras, O., Puaux, D., Gervais, R., & Ferry, B. (2016). Rigorous Training of Dogs Leads to High Accuracy in Human Scent Matching-To-Sample Performance. PLoS ONE, 11(2), e0146963. https://doi.org/10.1371/journal.pone.0146963

Pinc, L., Bartoš, L., Reslová, A., & Kotrba, R. (2011). Dogs Discriminate Identical Twins. PLoS ONE, 6(6), e20704. https://doi.org/10.1371/journal.pone.0020704

Prada, P. A., Curran, A. M., & Furton, K. G. (2011). The Evaluation of Human Hand Odor Volatiles on Various Textiles: A Comparison Between Contact and Noncontact Sampling Methods*,†. Journal of Forensic Sciences, 56(4), 866–881. https://doi.org/10.1111/j.1556-4029.2011.01762.x

Settle, R. H., Sommerville, B. A., McCormick, J., & Broom, D. M. (1994). Human scent matching using specially trained dogs. Animal Behaviour, 48(6), 1443–1448.

von Dürckheim, K. E. M., Hoffman, L. C., Leslie, A., Hensman, M. C., Hensman, S., Schultz, K., & Lee, S. (2018). African elephants (Loxodonta africana) display remarkable olfactory acuity in human scent matching to sample performance. Applied Animal Behaviour Science, 200, 123–129. https://doi.org/10.1016/j.applanim.2017.12.004

Wells, D., & Hepper, P. (2003). Directional tracking in the domestic dog, Canis familiaris. APPLIED ANIMAL BEHAVIOUR SCIENCE, 84(4), 297–305. https://doi.org/10.1016/j.applanim.2003.08.009

---

## [Decision Letter · Decision Letter 1]

6 Feb 2024

Development of an Automated Human Scent Olfactometer and its Use to Evaluate Detection Dog Perception of Human Scent

PONE-D-23-25721R1

Dear Dr. Aviles-Ros,

We’re pleased to inform you that your manuscript has been judged scientifically suitable for publication and will be formally accepted for publication once it meets all outstanding technical requirements.

Kind regards,

Tommaso Lomonaco, Ph.D

Academic Editor

PLOS ONE

Additional Editor Comments (optional):

Reviewers' comments:

Reviewer's Responses to Questions

**Comments to the Author**

1. If the authors have adequately addressed your comments raised in a previous round of review and you feel that this manuscript is now acceptable for publication, you may indicate that here to bypass the “Comments to the Author” section, enter your conflict of interest statement in the “Confidential to Editor” section, and submit your "Accept" recommendation.

Reviewer #1: All comments have been addressed

2. Is the manuscript technically sound, and do the data support the conclusions?

Reviewer #1: Yes

3. Has the statistical analysis been performed appropriately and rigorously? 

Reviewer #1: Yes

4. Have the authors made all data underlying the findings in their manuscript fully available?

Reviewer #1: Yes

5. Is the manuscript presented in an intelligible fashion and written in standard English?

Reviewer #1: Yes

6. Review Comments to the Author

Reviewer #1: Thank you for revising the manuscript following my comments. I did not find any concerns about dual publication, research ethics or so.

7. PLOS authors have the option to publish the peer review history of their article (what does this mean?). If published, this will include your full peer review and any attached files.

Reviewer #1: No

---

## [Editor Report · Acceptance letter]

22 Feb 2024

PONE-D-23-25721R1 

PLOS ONE

Dear Dr. Aviles-Rosa, 

I'm pleased to inform you that your manuscript has been deemed suitable for publication in PLOS ONE. Congratulations! Your manuscript is now being handed over to our production team.

Kind regards, 

on behalf of

Dr. Tommaso Lomonaco 

Academic Editor

PLOS ONE